# Melatonin Interplay in Physiology and Disease—The Fountain of Eternal Youth Revisited

**DOI:** 10.3390/biom15050682

**Published:** 2025-05-08

**Authors:** Cándido Ortiz-Placín, Ginés María Salido, Antonio González

**Affiliations:** Institute of Molecular Pathology Biomarkers, University of Extremadura, Avenida de las Ciencias s/n, E-10003 Caceres, Spain; coplacin@unex.es (C.O.-P.); gsalido@unex.es (G.M.S.)

**Keywords:** melatonin, health, disease, eternal youth, oxidative stress, cancer, aging, neurodegenerative diseases

## Abstract

Melatonin (N-acetyl-5-methoxytryptamine) is a hormone associated with the regulation of biological rhythms. The indoleamine is secreted by the pineal gland during the night, following a circadian rhythm. The highest plasmatic levels are reached during the night, whereas the lowest levels are achieved during the day. In addition to the pineal gland, other organs and tissues also produce melatonin, like, for example, the retina, Harderian glands, gut, ovaries, testes, skin, leukocytes, or bone marrow. The list of organs is extensive, including the cerebellum, airway epithelium, liver, kidney, adrenals, thymus, thyroid, pancreas, carotid body, placenta, and endometrium. At all these locations, the availability of melatonin is intended for local use. Interestingly, a decline of the circadian amplitude of the melatonin secretion occurs in old subjects in comparison to that found in younger subjects. Moreover, genetic and environmental factors are the primary causes of diseases, and oxidative stress is a key contributor to most pathologies. Numerous studies exist that show interesting effects of melatonin in different models of disease. Impairment in its secretion might have deleterious consequences for cellular physiology. In this regard, melatonin is a natural compound that is a carrier of a not yet completely known potential that deserves consideration. Thus, melatonin has emerged as a helpful ally that could be considered as a guard with powerful tools to orchestrate homeostasis in the body, majorly based on its antioxidant effects. In this review, we provide an overview of the widespread actions of melatonin against diseases preferentially affecting the elderly.

## 1. Introduction

Over a fairly long period of time, society has been subjected to insane threads that affect health and well-being and underlie the generation of diseases. In particular, the prevalence of certain diseases appearing in old age is increasing day by day. Genetic and environmental factors are majorly responsible for oxidative stress, which is a common condition related to most diseases. In this way, it is conceivable that treatment and, preferentially, disease prevention rank first on the podium as a challenge for society. It is common knowledge that diseases shorten life expectancy and profoundly affect quality of life. At present, special diets, intermittent fasting, calorie restriction, and the use of supplements, especially antioxidants, have emerged as valuable tools that are being used widespread in the fight against, and/or the prevention of, disease and to, expectedly, prolong life expectancy. In this regard, melatonin is a natural compound that is carrier of a not yet completely known potential that deserves consideration.

Oxidative stress is a condition caused by the accumulation of reactive oxygen species (ROS). An immediate consequence is damage to lipids, proteins, nucleic acids, and organelles, thus leading to the disruption of cell physiology [1]. If it is not controlled by antioxidant defense systems, it can create conditions that might lead to the development of disease [2]. Additionally, oxidative stress has been related to the occurrence of cellular senescence, which is a major mechanism mediating aging [1]. It has been suggested that long-lived individuals exhibit lower oxidative damage, indicated by the detection of lower plasma lipid peroxidation biomarkers [3]. As such, the ability of organisms to respond to oxidative stress is intricately connected to aging and life span [4]. Additionally, an important environmental risk factor that has been related to the development of disease in our actual society is referred to as bad habits, which include alcoholism, smoking, junk food, lack of movement (low exercise/sport), and negative emotions [5]. As such, bad habits create an imbalance of the body’s physiology that, in the long run, although it is also possible to occur in the short term, will unavoidably lead to the appearance of disease [6,7,8].

How we can prolong our life expectancy and prevent disease is a matter of constant research. Nevertheless, despite the infructuous search for health and longevity out there, it might be possible that, at least in part, the answer is much closer than we think.

Melatonin is a physiologically present indoleamine that is synthesized in the brain. It depicts widespread effects on all the organs and tissues of the body. Interestingly, melatonin exhibits the highest plasmatic levels at a young age, whereas its levels drop to a minimum in elderly, the latter being a period of life in which the body tends to break down, and diseases (cancer, neurodegeneration, chronic inflammation…) are more prone to appear [9,10,11,12,13]. Regarding oxidative stress, melatonin intake has a significant impact on improving oxidation parameters. Its beneficial actions are related to its antioxidant effects. As such, reduction in the levels of malondialdehyde (MDA) and of protein carbonyls (PCOs) and increased total antioxidant capacity (TAC) were noted in melatonin administered individuals [14,15].

Numerous studies exist that show interesting effects of the indoleamine in different models of disease. In this line, melatonin has emerged as a helpful ally with powerful tools to orchestrate homeostasis in the body. Most of the studies have been conducted in vitro. However, in vivo studies also exist, including studies in humans. The European Union Clinical Trials Register allows us to search for protocol and results related to melatonin use in clinical trials, which have yielded positive outcomes. Although most studies are focused on sleep disorders, melatonin has emerged as putative treatment, or at least a tool to improve symptoms, for cardiometabolic risk, sepsis, surgery, COVID-19, chronic pain, or damage by sun exposition. Roles as a neuroprotectant in infants, as an anxiolytic, or against visual anomalies have also been reported [16].

In this review, we describe the major functions that melatonin exerts in the body. We further hypothesize that indoleamine plays a protective role, potentially contributing to greater disease resistance, which keeps us healthy, mirroring the conditions observed in younger individuals. With the information given in this manuscript, we are not confirming that we have finally found “the fountain of eternal youth”. Notwithstanding, we would like to call the reader’s attention onto the fact that melatonin could be considered as a carrier of a not yet completely known potential, to a certain extent related to health and to life expectancy, that should be taken into consideration. Finally, the reader’s attention could be drawn to a recommendation that we have heard quite a few times: to prevent disorders, it is good to follow healthy habits. Additionally, we could dare to advise to “take a good sleep at the proper time” [17].

## 2. Melatonin and Its Physiological Functions

Melatonin (N-acetyl-5-methoxytryptamine) is a hormone associated with the regulation of biological rhythms. It was first isolated by Aaron B. Lerner in 1960 from bovine pineal glands [18]. The indoleamine is secreted by the pineal gland during the night, following a circadian rhythm. The pineal gland in humans is a small organ that weighs around 100–150 mg and presents a high vascularization. It is placed in the “geographical middle” of the brain, devoid of the blood–brain barrier and is attached to the roof of the third ventricle [19]. The highest plasmatic levels are reached during the night, whereas its levels are low during the day [20]. The concentration of melatonin amounts to 0–20 pg/mL in the daytime but reaches 40–100 pg/mL at night [21]. Interestingly, melatonin is a ubiquitous molecule within the body. In addition to the pineal gland, other organs and tissues also produce melatonin, such as, for example, retina [22], Harderian glands [23], gut [24], ovary [25], testes [26], skin [27], leukocytes [28], or bone marrow [29]. The list of organs is extensive, further including cerebellum, airway epithelium, liver, kidney, adrenals, thymus, thyroid, pancreas, carotid body, placenta, and endometrium [30]. At all these locations, the availability of melatonin is intended for local use. However, it is difficult to determine the exact concentration of melatonin that is locally released at the periphery organs. In general, it is accepted that the concentration of the indoleamine in the periphery is mostly higher than that found in blood. For example, the levels measured of melatonin in the gastrointestinal tissues account for 10–100 times that measured in blood [31]. Additionally, there is at least 400 times more melatonin in the gastrointestinal tract than in the pineal gland [32]. The concentration of the indoleamine in various parts of the gastrointestinal tract might vary among the distinct species. Notwithstanding, it reaches levels that are higher in comparison to those found in blood [24]. In testes, a study conducted by Kozioł et al. showed that melatonin concentration exhibited variations that depended on the stage of the year. For example, the levels were higher in May (522.50 ± 54.20 pg/mL) compared to July/August (258.50 ± 36.82 pg/mL). During September, the melatonin concentration was higher (393.50 ± 36.77 pg/mL) than in July/August but lower than in May [33]. Levels of 20 pg/mL of melatonin have been reported in the fluid of small follicles and 10 pg/mL in that of large ones [34]. In bone marrow, concentrations around 413 ± 81 pg/mg have been detected [35].

It has been reported that the oral administration of melatonin yields physiological levels in the blood of treated patients. A study conducted by Abdellah et al. showed that, up to 6 h, after the intake of tablets containing 1.9 mg of melatonin (a concentration much higher than the physiological concentrations reported for plasma), melatonin reached a concentration with physiological meaning (i.e., more than 100 pg/mL) [36]. Similar observations were reported by Aldhous et al., who administered tablets containing 2 mg of melatonin [37]. However, studies exist that signal that the bioavailability of oral melatonin ranges from 3 to 15% and that melatonin doses should be augmented depending on individual conditions, if the corresponding melatonin plasma levels are intended to be achieved [38,39,40].

Bacteria, protozoa, fungi, and invertebrates also produce melatonin [23]. Melatonin can also be found in plant food. To cite some, the indoleamine is present in cherries [41], bananas [42], pineapples [43], grapes [42], mangoes [43], nuts [44], tomatoes [45], oats [46], and mushrooms [44] (Table 1).

Melatonin exerts its effects majorly through the activation of two types of cellular membrane-located receptors, melatonin receptor type 1 (MT1) and 2 (MT2), which are widespread all through the body [47]. These membrane-bound receptors are coupled to G proteins and have been cloned in mammals [48]. Other cellular putative receptors signal the enzyme quinone reductase 2, which was termed the MT3 receptor [49], and the nucleus, where the termed retinoic acid-related orphan receptors (RORs) are located [50]. However, controversy exists regarding the existence and effectiveness of nuclear receptors for melatonin [50]. Receptor-independent effects have also been described [51].

In mammals, the synthesis of melatonin starts with the conversion of the amino acid tryptophan to 5-hydroxytryptophan by the enzyme tryptophan hydroxylase. Next, 5-hydroxytryptophan is converted into serotonin by the enzyme tryptophan carboxylase. This is considered the main precursor of melatonin synthesis. Afterwards, the enzyme aralkylamine N-acetyltransferase yields N-acetyl serotonin from serotonin. In a final step, the enzyme N-acetyl-serotonin methyltransferase generates melatonin as a final product [52] (Figure 1). Melatonin is transformed into metabolites, which include N-acetylserotonin, 5-methoxytryptamine, and some of its derivatives, and especially the 5-methoxylated kynuramines [53]. Many of them still possess biological or pharmacological properties [54].

The regulation of melatonin secretion follows a circadian cycle, which depends on an indirect pathway that originates from the retina, where photosensitive ganglion cells are activated by light. From there, a pathway reaches suprachiasmatic nucleus in the hypothalamus, via the retinohypothalamic tract. The role of the pineal gland is thus to receive and convey information about the light–dark cyclic variations in the environment. This is transformed into the cyclic production and secretion of melatonin, which reaches the highest plasmatic levels at night. The pineal gland therefore informs the body about the dark period, naturally occurring at the sleep phase, which is used to control and/or regulate physiological systems [19]. However, the absence of day-to-night variations in extrapineal melatonin concentration has been detected. This could be explained based on the existing signaling pathway(s), other than the photoperiod, which regulates the production of the indoleamine at these levels. Furthermore, it is accepted that locally produced melatonin out from the pineal gland is not released into the blood [55].

Melatonin exhibits as a major synchronization role of central but also peripheral oscillators (fetal adrenal gland, pancreas, liver, kidney, heart, lung, fat, gut, etc.), allowing the temporal organization of biological functions through circadian rhythms (24 h cycles) [56]. The indoleamine is also effective in the regulation of circadian rhythms, sleep [57], and reproduction [58]. Additionally, melatonin has shown additional, pleiotropic effects in the body. Melatonin exhibits anti-inflammatory action [59], antitumor effects [60], angiogenesis modulation [61], cellular differentiation regulators [62], telomerase activity modulation [63], immune system activation [64], skin/hair follicle protection [65], antioxidant effects and antioxidant system modulation [66], and the improvement of aging/frailty [67] (Table 1).

It goes without saying that impairment of the sleep/wake cycle will unavoidably interfere with melatonin secretion and its regulatory role [68]. In fact, the impairment of the retina-suprachiasmatic nucleus connection, as it occurs in blind people, will lead to the deregulation of melatonin homeostasis. The absence of negative feedback of light reaching the suprachiasmatic nucleus via the optic nerve can thus create a “free-running” rhythm of melatonin secretion [69]. Moreover, evidence suggests that melatonin secretion exhibits a decrease with age. In this line, plasmatic levels of melatonin drop significantly in the elderly in comparison with the values achieved in young people. The values of plasma melatonin are stable until the age of 30s–40s. Thereafter, a decline is observed [70]. As a consequence, low levels of melatonin might render the body more prone to the appearance of disorders [71]. Interestingly, strong reductions in circulating melatonin have been observed in disorders and/or diseases such as Alzheimer’s disease and other neurological and stressful conditions, pain, cardiovascular diseases, cancer, and/or endocrine and metabolic disorders, in particular, diabetes type 2 [72]. Therefore, it could be conceivable that melatonin supplementation might exert beneficial effects. It is accepted that supplementation with the indoleamine exhibits higher efficacy when endogenous melatonin levels are low. Additionally, most studies suggest that melatonin supplementation does not suppress endogenous production even with long-term use [73]. The study conducted by Zhdanova et al. reported increases in serum melatonin levels following a low oral dose of the hormone in elderly adults. Nevertheless, the peak reached exhibited variations among people over 48 years old [74]. Moreover, low-to-moderate dosages of melatonin (approximately 5–6 mg daily or less) appear safe, thus suggesting that long-term usage might be beneficial to certain patient populations [75].

Because of the widespread distribution of melatonin receptors and its variety of effects, a probable role for the indoleamine as a surveillant of body health status has been suggested [76]. It would be expected that the melatonin peak at night could convey photoperiodic information to every cell for chronobiotic synchronization, whereas that produced in extrapineal tissues and organs could serve as a protective mechanism against disease-related oxidants generated by normal and/or excessive activity [55]. However, the effects of melatonin seem to be cell-type and context-dependent. Moreover, because of the ubiquitous nature of melatonin receptors, it functions as a pleiotropic molecule. Furthermore, its multiplicity of actions goes beyond the established antioxidant activities. Interestingly, it protects healthy cells against noxious agents, whereas cell death is evoked in malignant or transformed cells [77,78].

## 3. Antioxidant Effects of Melatonin

As mentioned above, oxidative stress is critically responsible for the onset of disease and for the shortening of life expectancy. Moreover, evidence signals that melatonin’s effects are majorly due to its antioxidant actions. Melatonin itself is a free radical scavenging compound [79]. The indole scavenges a variety of oxygen and nitrogen-related radical species like hydroxyl radicals, hydrogen peroxide, singlet oxygen, nitric oxide, and peroxynitrite anions [80,81,82]. Its role as a free radical scavenger and antioxidant at both physiological and pharmacological concentrations has been observed in vivo [83]. However, controversy exists about the effectiveness in the antioxidant defense of radicals scavenging by nutritional antioxidants. Instead, other mechanisms such as enzymatic removal are rather the paramount antioxidant mechanisms [84,85]. Additionally, the expression of major antioxidant enzymes superoxide dismutase (SOD), glutathione peroxidase (GP), hemoxygenase 1 (HO-1), and NADPH/quinone oxidoreductase (NQO1), which are regulated by nuclear factor erythroid 2-related factor (Nrf2), is increased in the presence of melatonin [66]. Therefore, melatonin arises as a major protector against oxidative stress due to its regulatory role of the antioxidant defenses in different cell types and tissues. Next, in this section, we provide evidence for the antioxidant effects of melatonin and for the potentiation of the antioxidant response in different tissues and organs.

Melatonin induces the activation of Nrf2 and the antioxidant-responsive element (ARE) in pancreatic acinar cells [86]. Additionally, melatonin decreases fibrosis in the exocrine pancreas via modulation of the oxidative status in pancreatic stellate cells [87,88,89]. Melatonin protects cardiomyocytes against ischemia–reperfusion injury. Signaling pathways that were involved include Janus kinase 2/signal transducers and activators of transcription 3, nitric oxide synthase, and Nrf2 [90]. Neuroprotective effects of melatonin have also been shown. The indole normalized acrylamide-evoked changes in brain lipid content, as well as DNA oxidative damage and reduced glutathione levels [91]. Additionally, melatonin administration prevented neuronal damage due to lipid peroxidation. Moreover, the indole regulated energy metabolism and protected synaptic vesicle proteins from sepsis [92]. In an in vivo study, it was shown that melatonin protected lungs against lipopolysaccharide-induced ALI and pyroptosis by inhibiting the NLRP3-GSDMD pathway. Activation of the Nrf2/HO-1 signaling axis was observed [93]. Inflammation, oxidative changes, and the increase in autophagy were modulated by melatonin, thereby exhibiting protective action in hepatocytes. The indole also alleviated epithelial–mesenchymal transition (EMT) and pro-fibrotic changes [94]. Moreover, melatonin also prevents liver steatosis and exerts beneficial actions in organ transplantation and in ischemic reperfusion models [95]. Furthermore, melatonin modulates cirrhosis progression by inhibiting oxidation, inflammation, hepatic stellate cell proliferation, and hepatocyte apoptosis, thereby protecting the liver [96]. In the kidney, melatonin attenuated sepsis-induced acute injury by promoting mitophagy through SIRT3-mediated TFAM deacetylation [97]. Renoprotective effects of melatonin in obese and diabetic conditions have also been suggested [98]. Another study showed that melatonin exhibits protective effects against gemcitabine- and cisplatin-induced kidney injury. In the presence of melatonin, serum creatinine levels in the group treated with gemcitabine plus cisplatin treatment were normalized [99]. Colon injury induced by bisphenol A was attenuated by melatonin. The indoleamine targeted mitochondrial dynamics and the Nrf2 antioxidant system, and it activated the SIRT1/PGC-1α signaling pathway [100]. In the intestine, melatonin exerts potent anti-inflammatory action in acute trinitrobenzene sulfonic acid-induced colitis, presumably through its antioxidant action [101]. Further, melatonin decreased the progression of colitis-associated colon carcinogenesis via the downregulation of autophagy. The expression pattern of various autophagy markers such as Beclin-1, the LC3B-II/LC3B-I ratio, and p62 was diminished by treatment with the indoleamine [102]. Melatonin protected endothelial cells by decreasing reactive oxygen species (ROS) generation and lipid peroxidation levels. An increase in cell migration, the downregulation of pro-apoptotic proteins Cas 3, Cas 9, Cyt C, and Bax, and the upregulation of anti-apoptotic protein Bcl 2 were also noted [103]. Melatonin also improved vascular function in experimental hypertension. Reduction in intimal infiltration and the restoring of nitric oxide production were noted [104]. Another study showed that melatonin alleviated myocardial dysfunction through the inhibition of endothelial-to-mesenchymal transition via the NF-κB pathway. The indoleamine therefore attenuated endothelial cell dysfunction and ameliorated cardiac damage [105]. In striated muscle, melatonin improved restoration from cellular injury. The indoleamine rescued skeletal muscle differentiation and the melatonin/Pax7 axis. Melatonin could thus serve as a therapeutic agent to optimize muscle healing after injury [106]. The indoleamine also attenuated the expression of the fibrogenic cytokine, transforming growth factor β1, and reduced the phosphorylation of its downstream targets Smad2/3 both in vivo and in vitro, thereby exhibiting capacity to counteract muscle decline and fibrogenic conversion [107]. Again, melatonin effects are related to its antioxidant modulatory role. A study based on resistance-training athletes showed that melatonin treatment prevented extra- and intracellular oxidative stress induced by exercise and yielded skeletal muscle protection against exercise-induced oxidative damage [108]. In bone marrow, the application of melatonin partially restored the mitochondrial energy metabolism and osteogenic differentiation. The effects were explained by the restoration of mitochondrial redox homeostasis [109]. A further study revealed that melatonin displays the protection of bone marrow mesenchymal stem cells against iron overload-induced impairment of osteogenic differentiation and against senescence. The protective role involved blocking ROS accumulation and the inhibition of upregulation of p53, ERK, and p38 protein expression [110].

## 4. Melatonin and Neurological Diseases

Aging is a consequence of gradual and irreversible impairment of physiological processes. It is accompanied by a decline in tissue and cell functions that lead to potentially increased risks of developing various disorders, including neurodegenerative diseases, among others [111,112].

In a previous work, we reviewed the putative role of melatonin in the modulation of brain cell physiology and in the prevention of neurological diseases [17]. At this stage of this manuscript, we will focus onto the role of melatonin in Alzheimer’s and Parkinson’s disease and amyotrophic lateral sclerosis.

Alzheimer’s disease (AD) is a type of dementia that affects memory, thinking, and behavior. This is usually due to damages in the connections among neurons in parts of the brain involved in memory, including the entorhinal cortex and hippocampus. With time, it profoundly affects quality of life [113]. Consistent evidence suggests a putative role of melatonin in the prevention of AD development. Potentially, long-term melatonin treatment can protect AD transgenic mice against cognitive impairment and development of beta-amyloid (Aβ) neuropathology. Melatonin’s cognitive benefits could additionally involve its anti-inflammatory and/or antioxidant properties, which involved the expression of the antioxidant enzymes SOD-1, GP, and catalase [114]. Melatonin restored mitophagy, physiological process related to normal cellular function. The indoleamine improved mitophagosome–lysosome fusion via Mcoln1 and ameliorated mitochondrial functions, attenuated Aβ pathology, and improved cognition [115]. Another study revealed that melatonin alleviated hippocampus neurodegeneration and neuronal loss in an in vivo model of sleep-deprived rats. Lowering of the levels of inflammatory indicators such as interleukin (IL)-1β, IL-6, tumor necrosis factor (TNF)-α, inducible nitric oxide synthase (iNOS), and cycloxigenase-2 (COX2) were noted. Moreover, the treatment of rats with melatonin reversed the expression of Aβ42 protein [116]. Further evidence for a protective role of melatonin against AD derives from the study conducted by Chen et al., who showed that melatonin alleviated tau-related pathologies through the upregulation of miR-504-3p expression and targeting the p39/CDK5 axis [117]. To cite an additional study about the potential of melatonin in the prevention of AD, a study signaled melatonin as a candidate in arresting the intracellular accumulation of Aβ and protecting the cells from Aβ-related toxicity. The normalization of cell morphology and in the expression and phosphorylation of neurofilament proteins in wild-type murine neuroblastoma N2a were observed in the presence of melatonin [118].

Parkinson’s disease (PD) is a neurodegenerative disorder that predominantly affects the dopamine-producing neurons in an area of the brain termed the substantia nigra. A damaged brain leads to the involuntary shaking of parts of the body (tremor), slow movement, and stiff and inflexible muscles [119]. Disorder of the circadian rhythm, related to a decrease in circulating melatonin levels, has been considered a pathophysiological component of PD. Melatonin ameliorated neuroinflammation by inhibiting the polarization of microglia via a STAT-related pro-inflammatory pathway, therefore pointing out melatonin as an alternative option for neuroprotection in PD [120]. Additionally, potential correlations between the dopamine and melatonin serum levels and motor, cognitive, and sleep dysfunctions in patients with PD have been suggested [121]. Neuronal oxidative stress and mitochondrial dysfunction have been implicated in PD. Interestingly, melatonin slowed down the neurodegenerative process in a chronic mouse model of PD. Furthermore, defects of mitochondrial respiration, ATP, and antioxidant enzyme levels were normalized in the presence of the indoleamine [122]. Lipid peroxidation, concomitant with iron accumulation, leads to a process termed ferroptosis. This is a form of cell death that has been signaled to occur in the pathogenesis of PD. In the presence of melatonin, the activation of the Sirt1/Nrf2/Ho1/Gpx4 pathway was observed, which was related to a drop in the aggregation of α-synuclein (α-syn) and a decrease in ferroptosis. These findings highlight a neuroprotective role of melatonin in PD [123]. Moreover, the administration of melatonin to PD patients was effective in reducing the levels of oxidative stress markers. Decreases in the level of lipoperoxides, nitric oxide metabolites, and carbonyl groups in plasma samples from PD patients were detected. Additionally, increased catalase activity and improved mitochondrial activity were reported [124]. Another study showed increased levels of glutathione (GSH) and reduced oxidative stress after the administration of melatonin, which improved the symptoms of PD [125].

Amyotrophic lateral sclerosis (ALS) is a neurodegenerative disease that leads to motor dysfunction by both the damage of upper and lower motor neurons [126]. Evidence has shown that melatonin could inhibit the progression of ALS. In an in vivo model, the indoleamine reversed the upregulation of both SIRT1 and Beclin-1 expression and the LC3II/LC3I ratio in a dose-dependent manner and restored the ALS-induced downregulation of p62. Melatonin concentration dependently reversed the shortened ALS-induced survival time, as well as other parameters like weight loss and rotating rod latency decrease [127]. Melatonin protected cellular viability via the inhibition of Rip2/caspase-1 pathway activation, diminishing the release of mitochondrial cytochrome c and reducing the overexpression and activation of caspase-3 in a transgenic mouse model of ALS [128]. As in other tissues, the protective actions of melatonin are mediated through its antioxidant regulation function. This was shown in a study conducted in cultured motoneuronal cells, a genetic mouse model of ALS and in a group of patients with sporadic ALS. The indoleamine protected cultured cells against glutamate-induced toxicity, delayed disease progression, and the extended survival of transgenic mice and normalized to control values circulating serum protein carbonyls, a marker for oxidative stress, in ALS patients [129]. Finally, the study conducted by Bald et al. suggests that melatonin might slow the progression of ALS and prolong survival [130].

## 5. Melatonin and Cancer

The development of cancer may occur at any age, including infants. But cancer is mostly a disease of middle age and beyond. The incidence rates for cancer increase overall as age rises. The median age at diagnosis is 66 years old [131].

Melatonin has shown widespread action against a variety of cancer types, such as, for example, pancreatic, lung, liver, colon, breast, urinary bladder, skin, brain, gastric, prostate, kidney, bone, and/or leukemia. Here, we will mention some studies that report that melatonin might convey beneficial actions in cancer therapy. Apoptosis, uncontrolled cellular proliferation, invasion, and the metastasis of tumor cells are modulated by the indoleamine.

Studies conducted in the pancreatic cancer cellular model AR42J showed that melatonin modulated calcium (Ca^2+^) signaling, a major regulator of pancreatic cell physiology [132], and induced apoptosis [133]. A similar effect on apoptosis activation has been reported on the tumor cellular line PANC-1. Stimulatory effects by the indoleamine on Bcl-2/Bax and cas-9 protein expression were noted [134].

Lung cancer and the potential use of melatonin in treatments have also been studied. A trial found that a combined treatment of local radiofrequency ablation and melatonin inhibited the malignancy of non-ablated nodules and improved clinical outcomes for early lung cancer patients with multiple pulmonary nodules. Cotreatment depressed the activity of MAPK, NF-kappa B, Wnt, and Hedgehog pathways and upregulated P53 pathways. The combination also reversed the Warburg effect and decreased tumor malignancy [135]. Melatonin inhibited the proliferation of lung cancer cell lines A549, PC9, and LLC in vitro. The mechanisms of action of melatonin were related to cancer cell metabolism reprogramming, together with a shift from cytosolic aerobic glycolysis to oxidative phosphorylation [136].

In liver, melatonin depicted potent chemopreventive effects in inhibiting cholangiocarcinoma genesis and also reduced liver injury [137]. In hepatocellular carcinoma cells, melatonin suppressed mitochondrial respiration and glycolysis simultaneously, leading to anticancer effects [138]. Furthermore, melatonin modulated the motility and invasiveness of HepG2 cell in vitro. The molecular mechanism involved TIMP-1 upregulation and the attenuation of MMP-9 expression and activity. Inhibition of the NF-κB signal pathway was reported [139].

Regarding colon cancer, melatonin inhibited proliferation and viability and promoted apoptosis in colorectal cancer cells via upregulation of the microRNA-34a/449a [140]. In RKO colon cancer cells, melatonin might inhibit cellular migration by downregulating ROCK expression through modulation of the p38/mitogen-activated protein kinase signaling pathway [141]. Another study suggested that adding melatonin to standard irinotecan therapy might potentiate its anticancer effects in colon cancer treatment [142]. Similarly, co-treatment with 5-fluorouracil and melatonin suppressed tumor growth, proliferation, and tumor-mediated angiogenesis in colon cancer stem cells [143].

Melatonin exerts oncostatic effects on breast cancer via modulation of the immune and antioxidant responses. Based on the results available, as has been observed in studies on other types of cancer, melatonin arises as a promising candidate for combinatory use with conventional chemotherapeutics for breast cancer treatment. In this line, the combination of melatonin with doxorubicin reduced primary tumor growth and distant metastasis [144]. Moreover, the combination of thymoquinone and melatonin exhibited anticancer potential against breast cancer implanted in mice [145]. Another study showed that melatonin decreased metastasis, primary tumor growth, and angiogenesis in a mice model of breast cancer [146].

Urinary bladder urothelial carcinoma encompasses about 90% of all bladder cancer cases. Melatonin induced cell cycle arrest and suppressed tumor invasion in urinary bladder urothelial carcinoma. Downregulation of the HIF-1α and NF-κB pathways and downstream pathways, including Bcl-2, was observed [147]. Similar effects on cellular proliferation were reported by Wu et al. Their study reported that melatonin reduced proliferation and promoted the apoptosis of bladder cancer cells by suppressing O-GlcNAcylation of cyclin-dependent-like kinase 5 [148]. As reported above for other cancer types, the association of melatonin to chemotherapy drugs potentiates anticancer effects against urinary bladder cancer. Melatonin plus cisplatin suppressed bladder cancer cell growth/proliferation. The expression of proteins related to cell proliferation (PI3K/p-Akt/p-m-TOR/MMP-9/PrPC), cell cycle/mitophagy (cyclin-D1/clyclin-E1/ckd2/ckd4/PINK1), and cell stress (RAS/c-RAF/p-MEK1,2/p-ERK1,2) signaling were diminished by melatonin treatment. On the contrary, the expression of proteins related to apoptosis (Mit-Bax/cleaved-caspase-3/cleaved-PARP) and oxidative stress/mitochondrial damaged (NOX-1/NOX-2/cytosolic-cytochrome-C/p-DRP1) markers were increased [149].

Melanoma is an aggressive skin cancer originating from melanocytes. Anti-genotoxic and anti-mutagenic effects of melatonin supplementation were observed in a mouse model of skin cancer [150]. Melatonin reduced the growth of the human melanoma cells SK-MEL-1. The indoleamine induced the phosphorylation of p38 MAPK [151]. The antiproliferative and cytotoxic activity of melatonin in melanoma through a decrease in the activation of mitogen-activated protein kinase (MAPK) pathways was reported in a study conducted by Gatti et al. [152]. Another study, which was conducted employing melanotic (MNT-1) and amelanotic (A375, G361, Sk-Mel-28) melanoma cell lines, reported oncostatic responses and the control of mitochondrial function by melatonin. The uncoupling of oxidative phosphorylation, attenuation of glycolysis, dissipation of mitochondrial transmembrane potential, massive generation of ROS, and decrease in glucose uptake were observed [153]. Melatonin synergized the antitumor effects of vemurafenib in melanoma treatment. Enhancement of the inhibition of proliferation, colony formation, migration, and invasion was observed. Moreover, the association of the two compounds promoted vemurafenib-induced apoptosis, cell cycle arresting, and stemness weakening in melanoma cells [154].

Anticancer effects of melatonin in the nervous system have also been observed. Melatonin evoked changes in intracellular pH and metabolic modulation that diminished glioblastoma cell viability [155]. Agomelatine or ramelteon, agonists of melatonin receptors, have proven potential effects against glioblastoma via an increase in interleukin-2 synthesis, which is expected to reverse some of the immunosuppression associated with this type of cancer [156]. Another study revealed that melatonin inhibits tumorigenesis and the invasion of human glioblastoma, possibly by suppressing HIF1-α/VEGF/MMP9 signaling via the regulation of a variety of miRNAs [157]. Further research showed that melatonin blocked the expression of hypoxia-induced factor 1 alpha (HIF-1α) and that of downstream target genes, matrix metalloproteinase 2 (MMP-2), and the vascular endothelial growth factor (VEGF). Furthermore, melatonin destabilized hypoxia-induced HIF-1α protein via its antioxidant activity against ROS produced by glioblastoma cells in response to hypoxia. As a result, glioblastoma cell migration and invasion were suppressed [158]. Regarding the combination of melatonin with other treatments, evidence for a positive outcome exists. The combination of temozolomide and paclitaxel is the most used chemotherapy regimen for glioblastoma. The study conducted by Bostanci and Doganlar showed that melatonin reduced the viability of both glioblastoma and neuroblastoma cells. The indoleamine enhanced the cell cycle arrest and increased the expression of p53 and pro-apoptotic proteins (Bax and caspase-3), while it decreased the expression of anti-apoptotic protein Bcl-2 [159]. A recent study revealed that melatonin may amplify Nimotuzumab’s anti-glioma efficacy by inhibiting epidermal growth factor receptor (EGFR) dimerization [160].

Gastric cancer is another type of tumor in which melatonin has also shown beneficial effects. Melatonin induced apoptosis and inhibited the proliferation of SGC-7901 human gastric cancer cells via blockade of the AKT/MDM2 pathway. The indoleamine evoked cell cycle arrest and the downregulation of CDC25A, phospho-CDC25A (at Ser75), p21 (p21Cip1/p21Waf1), and phospho-p21 (at Thr145). Melatonin additionally induced the upregulation of Bax and the downregulation of Bcl-xL and increased the cleaved caspase-9 level and activated caspase-3 [161]. The indoleamine induced gastric cancer cells by suppressing cell proliferation and the induction of apoptosis via the regulation of PERK/eIF2α and HSF1/NF-κB signaling pathways [162]. A further study showed that the indoleamine inhibited the survival of human gastric cancer cells under endoplasmic reticulum stress involving autophagy and Ras-Raf-MAPK signaling. Increases in the expression of Bip, LC3-II, phospho-Erk1/2, and phospho-p38 MAPK were observed [163]. Further evidence on the antiproliferative actions of melatonin in gastric cancer cells was shown in the study conducted by Zhang et al., who reported that the indoleamine inhibited cell growth and migration and promoted apoptosis in gastric cancer cell line SGC7901 [164]. The combination of chemotherapy with melatonin augments the outcomes of cancer-directed drugs. Melatonin regulates cancer migration and stemness and enhances the antitumor effect of cisplatin [165] and 5-fluorouracil [166].

Prostate cancer is a common cancer among men. As in other cancer types, melatonin emerges as a compound with putative beneficial effects in its treatment. Melatonin suppressed androgen-dependent prostate cancer tumorigenesis [167]. The indoleamine modulated cell migration, cell invasion, cycle arrest in G0/G1 phase as well as the apoptosis of prostate cancer cells, as revealed in an RNA-seq study. Increase in the expression levels of 15-hydroxyprostaglandin dehydrogenase (HPGD), IL2Rβ, and nerve growth factor receptor (NGFR) were detected, whereas the expressions of insulin-like growth factor binding protein-3 (IGFBP3) and IL6 were diminished [168]. The study by Mayo et al. revealed that melatonin induced antitumor activity in prostate cancer via IGFBP3 and MAPK/ERK signaling and prolonged the survival of TRAMP mice by 33% when given at the beginning or at advanced stages of the tumor [169]. The indoleamine modulated metabolism for energy supply in prostate cancer cells via limiting glycolysis as well as the tricarboxylic acid cycle and pentose phosphate pathway [170]. Furthermore, melatonin exhibited antiproliferative effects against prostate cancer in vitro in culture and in vivo in the TRAMP model via the inhibition of Sirt1 [171]. Once more, combinations of anticancer treatments with melatonin improved the outcomes in prostate cancer therapy. A study revealed that melatonin increased the overall survival of prostate cancer patients with poor prognosis after combined hormone–radiation treatment [172].

Melatonin suppressed renal cell carcinoma progression through the ubiquitin/proteasome-mediated degradation of disintegrin and metalloprotease with thrombospondin motifs 1 (ADAMTS1). ADAMTS consists of a family of proteins widely implicated in tissue remodeling events undergoing in cancer development [173]. Most available studies on renal cancer signal the adjuvant role of melatonin when administered with chemotherapy drugs. In this line, melatonin improved the effects of sunitinib in suppressing Akt/mTOR/S6K activity, the induction of apoptosis, and the inhibition of cell growth in renal carcinoma cells via reversing the Warburg effect [174]. Furthermore, the indoleamine protects renal cells against toxicity induced by chemotherapy drugs such as docetaxel [175], 5-fluorouracil [176], or capecitabine [177].

Regarding bone cancer, around 0.2% of all malignancies are represented by bone sarcomas [178]. Melatonin interrupted osteoclast functioning and suppressed the tumor-secreted receptor activator of NF-κB ligand (RANKL) expression [179]. RANKL is an essential cytokine for osteoclast differentiation, induced by the metastatic tumor cells and responsible for the pathological bone resorption in bone metastasis [180]. A study conducted by Vimalraj et al. reported that the indoleamine suppressed tumor angiogenesis, modulating surrounding endothelial cell proliferation and migration, the morphology of blood vessels, and the release of angiogenic growth factors in osteosarcoma [181]. Melatonin was also suggested as a potentially useful and effective natural agent in the treatment of osteosarcoma in a study that showed inhibition of the biological functions of osteosarcoma cells via repression of the expression of lncRNA JPX. Regulation of the Wnt/β-catenin signaling pathway was involved [182]. Synergies between chemotherapy and melatonin have again been suggested in bone cancer. The combination of melatonin and zoledronic acid suppressed the giant cell tumor of bone in vitro and in vivo. The therapeutic effect might be achieved by inhibiting the activation of both the Hippo and NF-κB pathways [183]. Melatonin in combination with cisplatin increases the effectiveness of the latter in osteosarcoma cells [184]. Finally, melatonin induced osteoblast differentiation and bone formation via the transmembrane receptor MC3T3. This study suggested that the indoleamine promotes osteoblast differentiation and mineralization of the matrix and suggests an essential role in regulating bone growth [185].

Melatonin has also shown interesting effects against leukemia. Melatonin induced cytotoxicity in human leukemia cells, probably due to its pro-oxidant effect [186]. In human leukemia Molt-3 cells, melatonin evoked apoptosis through a caspase-dependent but ROS-independent mechanism. The study found that caspase-3, caspase-6, caspase-7, and caspase-9, but not caspase-8 and caspase-2, were quickly activated in the presence of melatonin. Upregulation of the pro-apoptotic factor Bax, with the release of cytochrome c from mitochondria, was also noted [187]. The indoleamine inhibited mixed lineage leukemia (MLL)-rearranged leukemia by suppressing the RBFOX3/hTERT and NF-κB/COX-2 signaling pathways. The blockade of NF-κB nuclear translocation and suppression of NF-κB binding to the COX-2 promoter and, thereby, suppression of the expression of COX-2 were observed [188]. Antileukemic effects of melatonin were also reported on wild-type and FLT3-ITD mutant cells. In this study, acute myeloid leukemia cells were employed. The indoleamine induced cell death, which was related to a decrease in glucose uptake, lactate dehydrogenase activity, lactate production, and HIF-1α activation [189]. Like in other cancer types, melatonin can enhance the effect of drugs used in the treatment of leukemia. Synergistical effects of a combination of melatonin with cytarabine and navitoclax contributed to the decrease in proliferative activity of leukemic cells. A decrease in the membrane potential of mitochondria and increase in the production of ROS and mobilization of cytosolic Ca^2+^ were noted [190]. Melatonin promoted puromycin-induced apoptosis involving the activation of caspase-3 and 5′-adenosine monophosphate-activated kinase-alpha in human leukemia HL-60 cells [191].

Finally, fibrosis is a condition that accompanies most cancers. Fibrosis is characterized by the activation of fibroblasts and immune cells, which contributes to the progressive deposition of extracellular matrix components and inflammation. Consequently, a termed tumor microenvironment is created, which helps tumor cells proliferate and resist treatments. Among other cellular types, cancer-associated fibroblasts (CAFs) and stellate cells contribute to cancer progression and the failure of treatments [192]. Melatonin has emerged as a potential agent that modulates fibrosis in different types of cancer, including pancreatic [193,194,195], gastric [196], liver [197], or lung [198] cancer, to cite some.

## 6. Melatonin and Immune-Related Diseases

Unresolved or continued oxidative stress can lead to inflammation. In turn, maintained inflammation can mediate the development of chronic diseases like cancer, diabetes, or cardiovascular, neurological, and pulmonary diseases [199]. The protective role of melatonin against inflammation has been extensively studied. Major attention has been paid to its antioxidant properties and to its potential to modulate antioxidant response in different cellular types, tissues, and organs, thus preventing or decreasing inflammation.

Melatonin prevented the development of oxidative stress and sustained the levels of GSH and GP activity in the pancreas of diabetic animals. Moreover, the indoleamine prevented the increase in pro-inflammatory cytokines and expression of Bax, caspase-3, and P53. Additionally, the anti-inflammatory cytokine IL-10 and anti-apoptotic protein Bcl-2 were increased in the presence of melatonin [200]. Activation of the melatonin receptor by agomelatine attenuated cadmium-induced oxidative stress and pancreatitis via the modulation of the Nrf2/HO-1 pathway. The analog ameliorated serum amylase and lipase levels, which were elevated by cadmium, as well as NF-kB p65, CD40, pro-inflammatory mediators, and caspase-3. Tissue injury was diminished, and antioxidant response was enhanced [201]. Finally, melatonin treatment exhibited beneficial effect on inflammation, apoptosis, and oxidative stress on the pancreas in a mouse model of senescence [202].

In hepatitis, the beneficial effects of melatonin could be related to the suppression of decompensation of the glutathione antioxidant system functions, recovery of liver redox status, and the attenuation of inhibition of the NADPH supply [203]. Melatonin increased glutathione concentration and activities of GP, glutathione reductase, NADP-isocitrate dehydrogenase, and glucose-6-phosphate dehydrogenase in the liver of rats undergoing toxic hepatitis [204]. Melatonin attenuated the extent of the damage caused in rabbits after experimental infection by rabbit hemorrhagic disease virus. A reduction in apoptotic liver damage, associated with the attenuation of endoplasmic reticulum stress, was observed upon treatment with the indoleamine [205]. The administration of melatonin to senescence-accelerated prone male mice decreased the mRNA expression of TNF-α, IL-1β, HO (HO-1 and HO-2), iNOS, MCP1, NFκB1, NFκB2, and NKAP. These effects were related to diminished inflammation of the liver in this mouse model [206].

Respiratory diseases are also mostly related to inflammation. In this regard, melatonin ameliorated lung cell inflammation and apoptosis caused by Klebsiella pneumoniae via AMP-activated protein kinase. This study was performed in vitro, employing lung cell lines HLF-1 and BEAS-2B that were infected with *K. pneumoniae*. Inflammation and apoptosis were observed, together with increased levels of IL-6, CXCL1, CXCL2, and caspase-9 mRNA. All these effects were abrogated by melatonin treatment [207]. The indoleamine also alleviated lung injury in influenza A virus H1N1-infected mice by mast cell inactivation and cytokine storm suppression [208]. Additional protective actions of indoleamine in lung were shown in the study by Ates et al., who reported that melatonin pretreatment modulated anti-inflammatory, antioxidant, YKL-40, and matrix metalloproteinases in endotoxemia rat lung tissue [209]. Lung ischemia–reperfusion injury derives from the production of ROS and the generation of inflammatory reaction. Melatonin inhibited oxidative stress, inflammation, and apoptosis and attenuated lung damage evoked by ischemia–reperfusion [210]. The mitochondrial quality control of A549 lung epithelial cells and primary alveolar type II cells was preserved through the SIRT3-dependent deacetylation of SOD2 in the presence of melatonin. SIRT3 further promoted the deacetylation of SOD2 at K122 and K68 [211].

Colon inflammation is characterized by disturbances in the intestinal microbiota and inflammation. Like in other tissues and organs, melatonin has been signaled to improve the resolution of colon inflammation. The indoleamine inhibited M1 macrophages, activated M2 macrophages, inhibited the secretion of pro-inflammatory factors, maintained colon homeostasis, and improved inflammation [212]. Melatonin supplementation normalized colitis, oxidative stress, mitochondria dysfunction, apoptosis, and the inflammation response induced by dextran sodium sulfate. Intestinal permeability and the level of IL-1β, TNF-α, iNOS, NLRP3, MDA, Bax, Cas-3, Cyt C, and Cas-9 were normalized by melatonin. The activation of the PI3K/AKT/Nrf2/RORα/SIRT1 pathway and suppression of NF-κB were detected [213]. It is also worth mentioning that the indoleamine mitigated radiation-induced gastrointestinal injury. Inflammation, villi shortening, apoptosis, and damage to goblet cells of the small intestine, in addition to moderate to severe inflammation, apoptosis, congestion, crypt injury, and goblet cell damage in the colon were observed following irradiation. It was concluded that the administration of melatonin after exposure to radiation may increase survival via the mitigation of damages to radiosensitive organs, including the gastrointestinal system [214].

Prostatitis is the most common urologic disease in adult males younger than 50 years. Oxidative stress plays a role in its development. If the causing agent or condition is not eliminated, the inflammatory process becomes chronic, and the progression of inflammation can lead to the possible development of prostate cancer. Antioxidants can therefore play an essential role in the treatment and/or prevention of prostate inflammation [215]. The study by Wang et al. demonstrated that melatonin inhibited the secretion of IL-1β, IL-6, and TNF-α in a chronic prostatitis model that employed lipopolysaccharide-treated RWPE-1 cells. The indoleamine alleviated inflammation and suppressed cell apoptosis and oxidative stress [216]. Similarly, melatonin attenuated prostatic inflammation and pelvic pain via the Sirt1-dependent inhibition of the NLRP3 inflammasome in an experimental autoimmune prostatitis mouse model [217]. Imbalance of the sexual steroid milieu and oxidative stress are often observed during aging and correlated to prostate disorders. A study developed in this line reported that melatonin triggered epithelial desquamation, reduced androgen receptor-positive cells, increased smooth muscle layer thickness, decreased corpora amylacea formation, and stimulated prostatic glutathione-S-transferase activity. The indoleamine thus partially recovered prostate damage and ameliorated degenerative alterations induced by aging [218].

Chronic interstitial nephritis has been associated with environmental and occupational exposure to glyphosate and hard water. Impairment of renal function might be attenuated by melatonin, possibly through the inhibition of ER stress and pyroptosis [219]. Melatonin depicted protective activity on lupus nephritis in mice. Its protective role was associated with the enhanced Nrf2 antioxidant signaling pathway and decreased renal NLRP3 inflammasome activation [220]. Melatonin upregulated heme oxygenase 1 (HO1) expression, diminished the production of ROS, reduced the expression of pro-inflammatory cytokines and increased the expression of anti-inflammatory cytokines, thereby ameliorating idiopathic membranous nephropathy, an autoimmune-mediated glomerulonephritis, in a murine model [221]. Finally, the indoleamine exerted protective effects in acute pyelonephritis, which could be ascribed to its ability to inhibit neutrophil infiltration, to balance the oxidant–antioxidant status, and to the modulation by the indoleamine of the generation of inflammatory mediators [222].

Inflammation affects immune surveillance and the responses to therapy [223]. The immune system protects the body from noxious agents such as bacteria, viruses, or fungi. Ordered/balanced immunity is crucial for maintaining health. Additionally, an insufficient level of immune defense leads to infections and tumors [224]. The unending lifestyle stressors along with genetic predisposition, environmental factors, and infections have set the immune system under a state of constant activation, leading to unresolved inflammation and increased vulnerability to chronic diseases [94]. Melatonin is considered a biological response modifier of the immune system with broad application in veterinary medicine. The indoleamine enhances a defined immune response in vivo [225]. Moreover, melatonin treatment promotes an increase in the weight of immune organs [226]. Studies exist that report favorable responses to vaccination, based on immune response improvement by the indoleamine. Melatonin enhanced vaccine-induced protective cellular immunity to HPV16-associated tumors [227]. Treatment with melatonin increased antibody titers 14 days after the immunization against the Venezuelan equine encephalomyelitis virus in mice. Increased antibody titters 14 days after the immunization were observed. IL-10 levels were also increased with melatonin treatment [228]. Treatment with melatonin limited the exacerbated local lung production of type I and type III interferons. This was probably associated with the observed improved symptoms in SARS-CoV-2-infected mice [229]. Primary Sjögren’s syndrome is an autoimmune disease that primarily affects exocrine glands. The administration of melatonin improved the hypofunction of the salivary glands, inhibited inflammatory development, and regulated clock gene expression in the animal model of primary Sjögren’s syndrome [230].

Various strategies for the treatment of cancer include surgical resection combined with chemotherapy, radiotherapy, nanotherapy, and immunotherapy. Studies have confirmed that melatonin mitigates the pathogenesis of cancer by directly affecting carcinogenesis [231]. Melatonin modulates macrophage polarization and prevents M2 induction. Additionally, the indoleamine prevents the conversion of fibroblasts into CAFs and prevents cancer cell stemness [232].

As shown above, the combination of melatonin with conventional drugs improves the drug sensitivity of cancers. Due to its modulatory effect on immune response, melatonin arises as an ally in immunotherapy [233]. The antitumor effect of melatonin in modulating the immunosuppressive tumor microenvironment by suppressing the YAP/PD-L1 axis was observed in non-small cell lung cancer [234]. In pancreatic adenocarcinoma, melatonin enhanced antitumor immunity through regulating tumor-associated neutrophils infiltration and NETosis, a program for the formation of neutrophil extracellular traps. Tumor cell apoptosis through cell-to-cell contact, fueled by fatty acid oxidation in neutrophils, was noted [235]. Another study revealed that melatonin inhibited epithelial–mesenchymal transition and downregulated PD-L1 expression in head and neck squamous cell carcinoma. Extracellular signal-regulated kinases/Fos-related antigen 1 (ERK1/2/FOSL1) pathways were involved, and synergistic effects with anti-PD-1 antibody were observed [236]. Melatonin regulated the levels of PD-L1 in macrophages via modulation of the associated microRNAs in the exosomes derived from gastric cancer cells. Treatment with melatonin increased the secretion of TNF-α and CXCL10 by the macrophages. Moreover, the recruitment of CD8+ T cells to the tumor site was noted, which resulted in the inhibition of tumor growth [237]. Another study signaled that melatonin-related long non-coding RNAs (lncRNAs) were potentially associated with a tumor immune microenvironment and might be therapeutic targets for breast cancer patients [238]. A recent study showed that melatonin suppressed the malignant features of lung cancer and enhanced treatment sensitivity by modulating the TME. The effect of the indoleamine was related to reversed EGFR-tyrosine kinase inhibitor (EGFR-TKI) resistance via the regulation of immune cell infiltration into the TME [239].

Last, but not least, melatonin can influence the success of organ transplantation. Melatonin administration in experimental models decreased rejection and improved transplant success [240]. Melatonin protected against oxidative stress during ovarian transplantation, thereby improving the outcomes [241]. Another study, conducted in ovariectomized mice subjected to heterotopic transplantation, revealed that melatonin protected from ischemic injury and reduced oxidative stress during the early days of transplantation [242]. In a study conducted employing a mouse model of type II diabetes mellitus, melatonin treatment improved human umbilical cord mesenchymal stem cell therapy. The indoleamine increased the p53-dependent expression of BCL2, inhibited BAX and Capase3 protein activation, and activated the phosphatidylinositol 3-kinase/protein kinase B (PI3K/AKT) response pathway [243]. After experimental kidney transplantation, melatonin protected kidney grafts from ischemia/reperfusion injury through the inhibition of NF-kB and apoptosis and improved survival. The induced tissue enzymatic activity of SOD simultaneous with a reduction in lipid hydroperoxide (LPO) was observed [244]. Additional research has shown that the inclusion of melatonin in preservation solutions prevented ischemic injury in rat kidneys [245] and increased the effectiveness of liver preservation solution [246]. Another study performed in this line revealed that exogenous melatonin enhanced bile flow and ATP levels after cold storage and reperfusion in rat liver. The indoleamine improved the restoration of liver function after cold storage and reperfusion [247].

## 7. Melatonin, Aging, and Frailty

The melatonin secretion level, comprising a prevalent nocturnal secretion persistence and the concomitant amplitude of the nocturnal peak, may be an important marker of biological age and of health status. An age-related decline of the circadian amplitude of the melatonin rhythm occurs in old subjects, especially in demented individuals [11]. The function of most organs in the body declines unavoidably throughout life, and concomitant senescence appears. Frailty is a common clinical syndrome in older adults that carries an increased risk for poor health outcomes, including falls, incident disability, hospitalization, and mortality [248]. Moreover, aging is associated with an increase in oxidative stress and inflammation [206]. The delay or moderation of these changes is a matter of increasing consideration nowadays. A range of factors can be used in the fight against aging. Dietary changes, weight control, practicing exercise, and the intake of various micronutrients are tools that are being used worldwide [67]. There is existent research signaling that melatonin could arise as a candidate that might help counteract the effects of aging in the body.

Melatonin and its metabolites target the prevention or reversal of skin aging [65]. Additionally, positive effects on age-induced cardiac functional and structural alterations have been proposed [249]. Furthermore, melatonin supplementation significantly improved in vitro fertilization success rates in women of advanced maternal age. In cumulus cells from patients, melatonin enhanced cellular resilience against oxidative stress and metal-induced toxicity. Reduced oxidative stress markers, improved mitochondrial function, and restored expressions of glycolysis and tricarboxylic acid cycle-related genes were observed [250]. Another study on the reproduction system of aged mice revealed that melatonin supplementation prevented testicular aging. Repair in seminiferous tubules and interstitial tissues with the enhancement of spermatogenesis was observed in melatonin-treated animals [251]. Melatonin exhibited a suppressive role in the aging process of colonic tissue via decreasing SIRT2 expression. Again, anti-oxidative stress actions of the indoleamine are involved [252]. In skeletal muscle, atrophy and fibrosis were counteracted by melatonin. The indoleamine restored the myogenic potential and inhibited the fibrogenic conversion of satellite cells, a process that plays a role in the pathogenesis of age-related sarcopenia. Melatonin treatment mitigated the loss of muscle mass and strength in aged mice, replenished the satellite cell pool and curtailed muscle fibrosis [107]. A study on kidney function revealed that tubular and glomerular structures and tissue antioxidant enzyme activities were well preserved in melatonin-administered rats. The indoleamine reduced tissue MDA levels, increased tissue SOD, CAT, and GP activities, and elevated GSH levels in old animals [253]. Additionally, a neuroprotective role for melatonin during aging has been suggested. As such, the indoleamine, as well as its metabolites, could help with retarding brain aging and, probably, the development of age-related neurodegenerative diseases, such as Alzheimer’s disease, Parkinson’s disease, Huntington’s disease, multiple sclerosis, or amyotrophic lateral sclerosis [254]. SAMP8 mice are used as a model of aging. The mice show early cognitive loss that mimics the deterioration of learning and memory in the elderly. Melatonin enhanced SIRT1 expression in neuron cultures obtained from these mice and diminished mitochondrial dysfunction and the subsequent increase in cellular oxidative stress [255].

Immunosenescence is referred to as a condition of destruction and remodeling of immune organ structure as well as innate and adaptive immune dysfunction that occurs with aging. This leads to poor vaccination outcomes and increased susceptibility to infection, age-related disease, and malignancies [256]. Melatonin could be a pharmacological candidate for aged and immuno-compromised individuals. In this line, it has been suggested that melatonin supplementation may enhance immunity in aged individuals by upregulating immunosenescence indices in association with T lymphocytes [257]. In the animal model Streptopelia risoria, melatonin and its precursor tryptophan and the amino acid increased cell viability and resistance to the induced oxidative stress of blood heterophils and enhanced phagocytic function. Reversion by melatonin of the immunosuppressor and oxidative effects accompanying phagocytosis at advanced ages is proposed [258]. Another study from this same group revealed that tryptophan enhanced heterophil viability, phagocytic response, and the detoxification of superoxide anion radicals, thereby supporting an immunoregulatory action of melatonin [259]. Melatonin modulated immunity in young-adult and aged squirrels. This was measured as the normalization of total leukocyte count, lymphocyte count, and the percentage stimulation ratio of splenocytes against T cell mitogen concanavalin A [260]. Finally, additional evidence that the treatment with melatonin could slow down the effects of the aging process of immune responses derives from the study conducted by Baeza et al., who showed improvement of leucocyte functions in ovariectomized aged rats after treatment with various hormones, including melatonin. Chemotaxis, the lymphoproliferative response to the mitogen concanavalin A (Con A), the release of IL-2 and the natural killer (NK) cell activity were partially or completely reversed by treatment with the indoleamine [261].

## 8. Melatonin and Healthy Aging: Impairment of Sleep–Wake Cycle and Disease

Nowadays, human life expectancy is prolonged in comparison to the past. A consequence is the thriving of age-related diseases. Aging can be considered a multifactorial process, which is complex. Molecular and cellular decline, apart from other yet unknown factors, contributes to the process. Altogether, aging affects tissue function over time, and the organism becomes frail and susceptible to disease and death. Slowing human aging in addition to extending molecular and physiological youthfulness, vitality, and health are major challenges that have been sought to be solved in the past and present. Calorie restriction has been signaled as a strategy for extending health and lifespan in most biological models assessed. Calorie restriction refers to diminishing average daily caloric intake below the typical or habitual amount, without malnutrition or the deprivation of essential nutrients [262]. Nevertheless, other tools might exist but, to date, are not completely known; these could contribute to healthy aging and the retardation of organism impairment.

Melatonin could be associated with key aspects of healthy aging and longevity. The indoleamine influences energy metabolism, autophagy, and circadian rhythms, thereby modulating aging and neurodegeneration. Insulin-like growth factor 1 (IGF-1), Forkhead box O (FoxOs), sirtuins, and the mammalian target of rapamycin (mTOR) signaling pathways are thought to be involved [263]. An interesting study suggested anti-aging actions of melatonin on lifespans, effects that were evaluated in *Caenorhabditis elegans*. Different dosages of melatonin affected the lifespan and morphology of the nematode. The authors concluded that the indoleamine might be used in the prevention of aging [264]. Ferrari et al. carried out a study on young controls, healthy old subjects, and centenarians. They observed that, only in centenarians and in young controls, the excretion of 6-hydroxymelatonin sulfate (aMT6s) was significantly higher at night than during the day. They also found that the circadian rhythm of melatonin secretion was maintained in centenarians and signaled that maintaining the secretion cycle and the plasma levels of melatonin could be considered one factor in successful aging [265]. In another study, urinary melatonin levels were used as a biomarker related to cognitive function, physical function, and mortality in older men. The authors suggested that circadian markers like melatonin might be associated with healthy aging and longevity. However, little evidence of associations was reported [266].

Alterations in the sleep–wake cycle, due to insomnia, night shift work, or other causes, produce an alteration in melatonin secretion and, therefore, could result in the disruption of the balance of cellular function. Body function deterioration, which underlies aging and frailty and leads to the development of diseases, is related to the onset of uncontrolled and/or maintained oxidative stress. Consequently, in the long term, health status may be compromised. It is well established that the probability of developing cancer is higher in individuals who work night shifts [267]. Furthermore, plasma melatonin levels are lower in elderly individuals, a stage of life in which the likelihood of developing cancer and neurodegenerative diseases is elevated [268]. Therefore, the preservation of a robust circadian rhythmicity (particularly related to the sleep/wake cycle) is crucial for healthy aging. Additionally, proper nutrition and adequate physical exercise are also important. Studies have revealed that aging comes along with circadian alteration, including disrupted sleep and inflammation, which leads to metabolic disorders. Thereby, sleep cycle disturbances cause numerous pathophysiological changes that could accelerate the aging process [269]. Thus, melatonin might serve as a whole-body protector against oxidative stress and, therefore, functions as a guard against health disturbing conditions (Figure 2).

It is well known that light suppresses melatonin production in the body, with consequent negative outcomes for sleep. Without melatonin, sleep onset will be impaired, or sleep will be more superficial, fragmented, and marked by increased awakenings. In summary, sleep quality will be compromised. Studies exist that signal that sleep–wake disturbance has been implicated in the pathogenesis of chronic liver disease [270], in AD pathogenesis [271], in gastrointestinal functioning and digestive diseases [272], in the impairment of thyroid function [273], in the pathogenesis muscular dystrophy [274], in cardio-metabolic diseases and impairment of eating patterns [275], and in metabolic disorders (including obesity) [276]. Therefore, it is understood that sleeping for a short time and/or with light instead of darkness will have undeniable consequences on health status and might affect healthy aging.

## 9. Conclusions

Bearing in mind that most diseases have an oxidative stress basis, the availability of antioxidants within the body can represent a powerful tool to prevent damage that, if not sorted out, could evolve towards diseases. Melatonin exhibits a wide variety of effects within the human body, controlling a plethora of functions. The indoleamine is mainly released during sleep at night. Thus, there is no wonder that having insufficient and poor-quality sleep, due to sleeping in the light or other reasons, can have undesirable effects. Additionally, blood levels of melatonin are lower in the elderly, a stage in life in which the probability of cancer and neurodegeneration development increases. Frailty also appears. Because the bioavailability of exogenous melatonin as a diet supplement is low, it could thus be expected that an adequate rhythm of melatonin secretion in the body appears to be a powerful tool to prevent damage to body tissues and structures. Evidence signals that only supplementation might represent a helping hand when physiological release is impaired. The information available reports that melatonin is safe to use long term.

Nevertheless, a certain deal of controversy exists. For instance, the subcellular localization of melatonin synthesis and the full spectrum of its molecular targets have not been entirely unraveled. Additionally, based on clinical reports, some doubt persists mainly concerning the many functions that have been reported for melatonin [277]. Therefore, further research is needed to clearly confirm the plethora of effects exerted by the indoleamine and, moreover, to establish the safety of exogenous melatonin for prolonged periods of time.

Last, but not least, at present, it is difficult to draw any definitive conclusions on the effectiveness of melatonin in reversing aging and restoring, or even prolonging, youth. Despite the research that has shed light on this topic, scientists still have not found compelling evidence that signals that melatonin supplementation is a valuable tool to prevent disease and/or enlengthen life expectancy. Additional research, involving clinical trials and humans exhibiting long life expectancy, will be needed to finally achieve this goal.

## Figures and Tables

**Figure 1 biomolecules-15-00682-f001:**
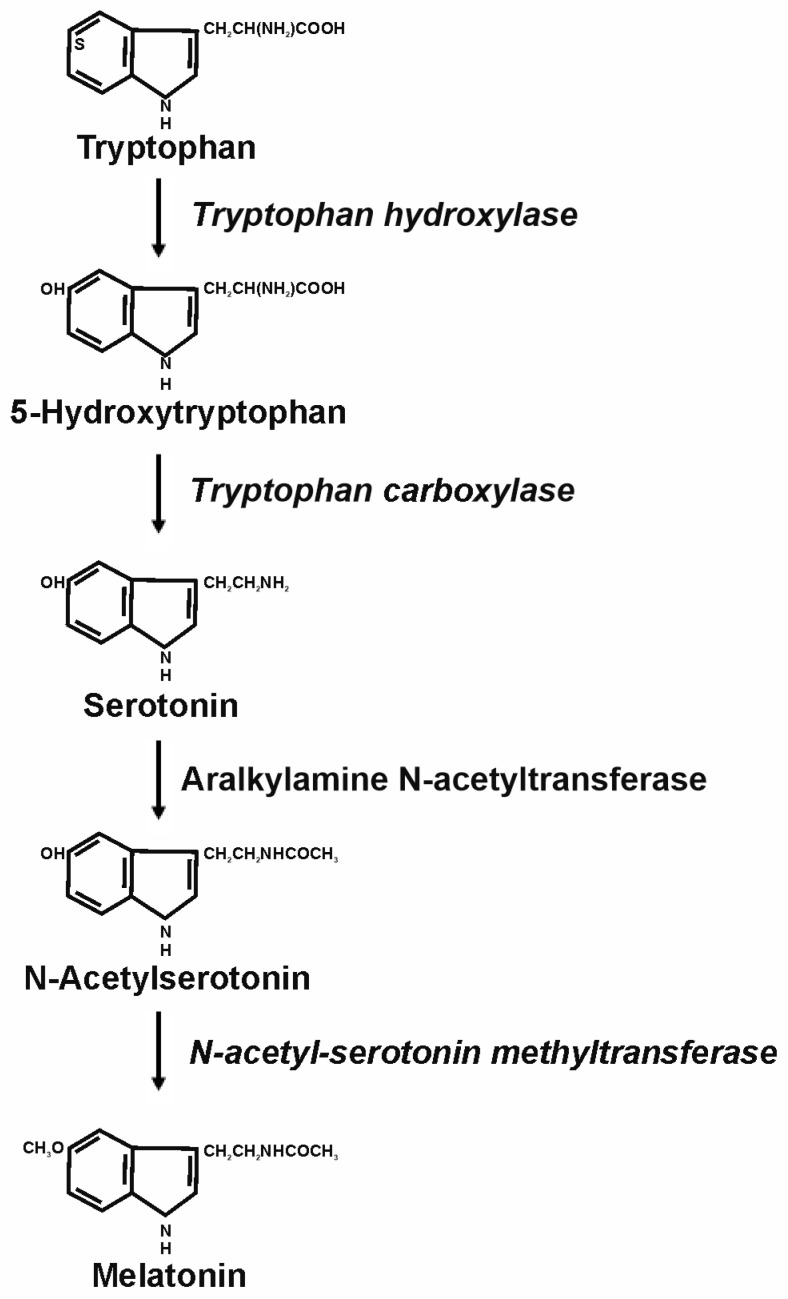
Pathway for melatonin synthesis in the body. The source for melatonin synthesis is the amino acid tryptophan, an essential amino acid that is not naturally present in the body and, therefore, must be taken with food. Tryptophan is transformed into melatonin by the action of various enzymes, following a sequential chain of reactions.

**Figure 2 biomolecules-15-00682-f002:**
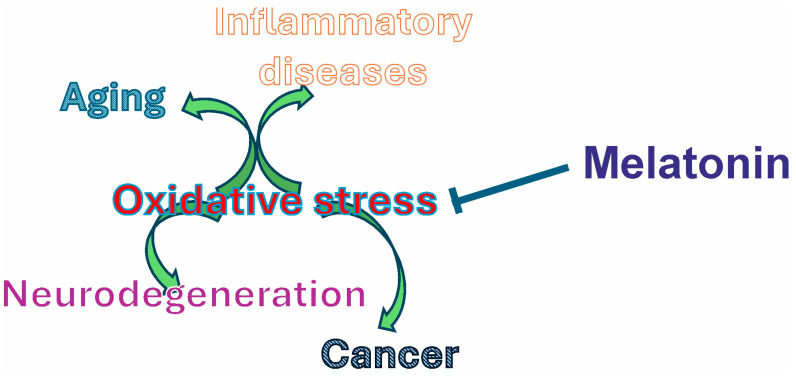
Scheme summarizing the relationship of oxidative stress with disease. Melatonin, due to its “antioxidant surveillance role”, might serve as a protector of health.

**Table 1 biomolecules-15-00682-t001:** Summary of the effects of melatonin in the body, and list of fruits and vegetables in which the indoleamine is present.

Effect	Fruit/Vegetable
Body clock synchronization	Cherries
Sleep	Bananas
Reproduction	Pineapples
Anti-inflammatory	Grapes
Antitumor	Mangoes
Cell differentiation	Nuts
Telomerase activity	Oats
Angiogenesis regulator	Tomatoes
Immune system activation	Mushrooms
Antioxidant	
Anti-aging	

Melatonin exhibits numerous effects on the organs and tissues of the body. Apart from the pineal gland and other organs, the indoleamine is present in nature, as it is found in fruits and vegetables. These can represent a valuable source of melatonin, which can enter the body via the digestive system (for more details, see reference list).

## Data Availability

No new data were created or analyzed in this study.

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
