# Peer review of "Melatonin Interplay in Physiology and Disease—The Fountain of Eternal Youth Revisited"

_biomolecules, 2025, doi:10.3390/biom15050682_

Round 1

Reviewer 1 Report

Comments and Suggestions for Authors

This article summarizes extensive literature on the role of melatonin in the pathogenesis of various human diseases. The review demonstrates considerable data synthesis and comprehensive coverage, but the logical structure and coherence remain unclear. Further revisions are recommended before considering acceptance. Detailed feedback for each section is provided below:

Abstract:

  • Lines 14-15: The statement could be improved as: "Genetic and environmental factors are the primary causes of diseases, with oxidative stress serving as a key contributor to most pathologies."

  • A concluding summary is needed in the abstract to explicitly outline the aspects of melatonin's roles reviewed in this article.

Section 1 (Lines 77-78):

  • The introduction of oxidative stress appears abrupt and lacks contextual linkage.

  • Expand this section to clarify the relationship between oxidative stress and longevity, as well as the connection between oxidative stress and melatonin.

Section 2 (Line 170):

  • Emphasize that the described melatonin synthesis pathway applies specifically to mammals.

Section 3:

  • Justify the rationale for focusing on oxidative stress in this context.

Section 4:

  • Add an introductory explanation of the relationship between neurodegenerative diseases and longevity.

Section 5:

  • Include the typical age range for cancer onset to better align with the overarching theme of longevity.

Sections 6-7:

  • These sections address melatonin's links to physiological processes (e.g., immunity), whereas other peer sections focus on diseases. It is recommended to merge Sections 6-7 into a single section titled "Melatonin and Immune-Related Diseases."

Section 10 (Conclusion):

  • The conclusion is overly lengthy and requires conciseness.

General Notes:

  • The overall structure and details appear disorganized in the revised manuscript, with some sections disrupted by partial edits. A thorough restructuring and refinement are strongly advised.

Comments on the Quality of English Language

I hope to revise the relevant structure more carefully

Author Response

Reply to Reviewer 1 Biomolecules-3549664

This article summarizes extensive literature on the role of melatonin in the pathogenesis of various human diseases. The review demonstrates considerable data synthesis and comprehensive coverage, but the logical structure and coherence remain unclear. Further revisions are recommended before considering acceptance. Detailed feedback for each section is provided below:

Thank you very much for reviewing our manuscript, for the time given to our work and for your suggestions, which we think have helped us improve the quality of our manuscript. Here we send our replies to your comments and suggestions.

Abstract:

Comment: Lines 14-15: The statement could be improved as: "Genetic and environmental factors are the primary causes of diseases, with oxidative stress serving as a key contributor to most pathologies."

A concluding summary is needed in the abstract to explicitly outline the aspects of melatonin's roles reviewed in this article.

Reply: The abstract has been rewritten. We have considered the reviewer´s suggestion and we have included the proposed statement in the text. Additionally, we have included a concluding summary at the end of the abstract.

Comment: Section 1 (Lines 77-78):

The introduction of oxidative stress appears abrupt and lacks contextual linkage.

Expand this section to clarify the relationship between oxidative stress and longevity, as well as the connection between oxidative stress and melatonin.

Reply: Following the reviewer´s suggestion we have expanded the beginning of this section, including detailed information about definition of oxidative stress, its relationship with aging and longevity and a connection of oxidative stress and melatonin. New bibliography has been cited and included in the references list.

The text included is the following: “Oxidative stress is a condition caused by the accumulation of reactive oxygen species (ROS). An immediate consequence is damage to lipids, proteins, nucleic acids and organelles, thus leading to the disruption of cell physiology [1].” … “Additionally, oxidative stress has been related to the occurrence of cellular senescence, which is a major mechanism mediating aging [1]. It has been suggested that long-lived individuals exhibit lower oxidative damage, indicated by detection of lower plasma lipid peroxidation biomarkers [3]. As such, ability of organisms to respond to oxidative stress, is intricately connected to ageing and life span [4].” Lines 56-65.

“Regarding oxidative stress, melatonin intake has a significant impact on improving oxidation parameters. Its beneficial actions are related to its antioxidant effects. As such, re-duction of the levels of malondialdehyde (MDA) and of protein carbonyls (PCO), and in-creased total antioxidant capacity (TAC) were noted in melatonin administered individuals [14,15]”. Lines 80-83.

Comment: Section 2 (Line 170):

Emphasize that the described melatonin synthesis pathway applies specifically to mammals.

Reply: Thank you very much for this comment. We have signaled that melatonin synthesis pathway applies to mammals. Line 167.

Comment: Section 3:

Justify the rationale for focusing on oxidative stress in this context.

Reply: We have added a short introductory paragraph to this section. In the revised manuscript we state “As mentioned above, oxidative stress is critically responsible for the onset of disease and for the shortening of life expectancy. Moreover, evidence signals that melatonin effects are majorly due to its antioxidant actions”. Lines 242-244.

Comment: Section 4:

Add an introductory explanation of the relationship between neurodegenerative diseases and longevity.

Reply: Following the reviewer´s suggestion we have included an introductory explanation of the relationship between neurodegenerative diseases and longevity. New bibliography has been cited and included in the references list.

The text is as follows “Aging is a consequence of gradual and irreversible impairment of physiological processes. It is accompanied by a decline in tissue and cell functions which, thus, lead to potentially increased risks of developing various disorders, including neurodegenerative diseases, among others [111,112]”. Lines 317-320.

Comment: Section 5:

Include the typical age range for cancer onset to better align with the overarching theme of longevity.

Reply: Thank you very much for this suggestion. We have included information regarding this comment in the text as follows: “Development of cancer may occur at any age, including infants. But cancer is mostly a disease of middle age and beyond. The incidence rates for cancer overall increases steadily as age rises. The median age at diagnosis is 66-year-olds [131].”. Lines 394-396.

Comment: Sections 6-7:

These sections address melatonin's links to physiological processes (e.g., immunity), whereas other peer sections focus on diseases. It is recommended to merge Sections 6-7 into a single section titled "Melatonin and Immune-Related Diseases."

Reply: Following the reviewer´s suggestion, sections 6 and 7 have been merged into a single section.

Comment: Section 10 (Conclusion):

The conclusion is overly lengthy and requires conciseness.

Reply: Following the reviewer´s suggestion, we have shortened the discussion.

Comment: General Notes:

The overall structure and details appear disorganized in the revised manuscript, with some sections disrupted by partial edits. A thorough restructuring and refinement are strongly advised.

Reply: Thank you very much for this comment. Restructuring of the manuscript has been carried out. Additionally, some sections were edited by the editorial office, we think, prior to submission to reviewers.

Reviewer 2 Report

Comments and Suggestions for Authors

see file

Author Response

Reply to Reviewer 2 Biomolecules-3549664

The paper of Ortiz-Placín et al deals with the “interplay” between melatonin with disease, essentially thanks to its antioxidant capacity.

Obviously, this essay – not a review – has been reviewed at least once before as revealed by the many red-corrected sentences. Furthermore, when those red sentences were replacing previous sentences, the old sentence(s) were not erased. Thus, fishing and guessing became one of the tasks I have to do to follow the text. Quite unfortunate.

This very fact does not make my job easier, as I will regard it with very fresh eyes, and maybe I will criticize new points that were not touched previously.

Thank you very much for reviewing our manuscript, for the time given to our work and for your suggestions, which we think have helped us improve the quality of our manuscript.

We would like to first start our report stating that the editorial office performed changes and corrections on the submitted version of the manuscript and might, thereafter, send the paper with “marked changes” to the reviewers. We agree that this might cause disturbance during the reviewing process. Therefore, we appreciate very much the effort made by the reviewer. Here we send our replies to your comments and suggestions.

I recommend the essay to be clearly stating at least 2 points – and maybe 3, that are NOT present in its present form:

Comment 1/ A discussion on concentration. Numerous (this is an understatement) preclinical studies show some effect at stratospheric concentrations. This should be pointed out. Strongly. Because this kind of concentrations cannot be reached in Human.

Reply: Thank you very much for this observation. In the first version of our manuscript, we mentioned that “the levels measured of melatonin in the gastrointestinal tissues account for 10-100 times that measured in blood. Additionally, there is at least 400 times more melatonin in the gastro-intestinal tract than in the pineal gland”, now lines 125-126.  New information regarding this comment has been included in the text, lines 122-125, which says “However, it is difficult to determine the exact concentration of melatonin that is locally released at the periphery organs. In general, it is accepted that the concentration of the indoleamine in the periphery is mostly higher than that found in blood”.

The reviewer is right in the line that patients have been receiving high concentrations of melatonin, mainly orally. We could find information about the concentration of melatonin that is reached in plasma after taking an oral dose of the indoleamine. This is now mentioned in the text and bibliography is cited.

Lines 138-143: “It has been reported that oral administration of melatonin yields physiological levels in blood of treated patients. The study by Abdellah et al. showed that up to 6 h after intake of tablets containing 1.9 mg melatonin (a concentration much higher than the physiological concentrations reported for plasma), melatonin reached a concentration with physiological meaning (i.e., more than 100 pg/mL) [36]. Similar observations were reported by Aldhous et al., who administered tablets containing 2 mg melatonin [37]”.

Comment 2/ Controversies papers have been published. On some key points. Privately, I know specialists (neuro and cancer) that are alarmed by some of the claimed made on those subjects. This must be emphasized.

Reply: We do not have clear what this comment refers to. If this refers to the exact concentration of melatonin that is reached at certain locations, we have disclosed it in the manuscript, lines 122-124: “However, it is difficult to determine the exact concentration of melatonin that is locally re-leased at the periphery organs”.

If the comment refers to the confirmation of melatonin effectiveness for the treatment of diseases as cancer or neurodegeneration, for example, we have included the following text in the Conclusion, lines 923-929: “Additionally, at the present, it is difficult to draw any definitive conclusions on the effectiveness of melatonin in reversing aging and restoring, or even prolonging, youth. Despite research seeds some light in this line, scientists still have not found compelling evidence that signals that melatonin supplementation is a valuable tool to prevent disease and/or enlengthen life expectancy. Additional research, involving clinical trials and humans exhibiting long life expectancy, will be needed to finally achieve this goal”.

Comment 3/ (Maybe), insisting on the fact that clinical studies (about 400 of them) failed in showing a melatonin effect on multiple diseases. This must be signified to the reader.

Reply: Please, see reply to comment 2.

Comment: The review is well written. Maybe a bit distant from the actual fact (“Melatonin-does- everything”) that should be at least surprise any scientists. Please state as well that the paper has not been generated, at least in part, by AI.

Reply: Thank you very much for your comment. We confirm that the paper has not been generated, either totally nor in part, by AI.

Title: I am not in favor of sensationalism, and thus, the allusion of the” fountain of eternal youth” seems to me disproportionate.

Reply: We appreciate the reviewer´s observation. We do not think we were being sensationalists when we wrote the title of our manuscript. Sensationalism would be rather to state “Melatonin, the fountain of eternal youth found at last” or “Melatonin is the revealed fountain of eternal youth”.

The term “fountain of eternal youth” has been used or mentioned in earlier studies; please, follow the link “https://pubmed.ncbi.nlm.nih.gov/?term=fountain+of+eternal+youth”.

Moreover, we state early, in the Introduction section, line 94, “We are not confirming in this manuscript that we have finally found the fountain of eternal youth”.

Abstract First, the abstract is not an abstract but an introduction explaining the many diseases of the Human race, and linking them – why not? – to oxidation. Although an interesting prospect, it does not summarize the view(s) of the paper.

Only the last sentence introduced the “victim of the day”, melatonin.

This suggests that melatonin can be used to prevent almost all diseases – at least those linked to oxidation. Hopefully, later in the text a list of such diseases is given.

That would be the task N°1: given the reader an abstract telling what the paper is all about. Melatonin, of course, but why.

Reply: Thank you very much for this observation. We have rewritten the abstract.

1/ Introduction.

The introduction seems to reveals to the incult (uncultivated) reader the many numbers of references in Human History on the search of eternal youth. It seems that a couple of biblical – and Coran’s – references are missing.

The second part of the introduction is about oxidative stress (lines 77 – 94). The usual suspect in Diseases. Although again, an interesting prospect, I do not think it is a fact, but rather an hypothesis that would require intensive research and critical complete review of the whole literature. Of course, one can also declare that oxygen is responsible, and taking off the picture would solve many problems….

Finally, the third part (Lines 95 and following) talks about melatonin. The key reference to melatonin action is an obscure book (sorry to the authors of this book, on Green Tea… Later on, the bibliographic list contains 2 references 5, I guess the second one is much more adequate ). That is not the main reference of melatonin n atu ral (normal?) action one could scientifically considered. See below.

Reply: We have removed the list of historical and artistic quotations from the introduction. Additionally, we have rewritten the section and included comments suggested by the other reviewers.

2/ Melatonin first paragraph/ Background.

After this brief introduction on melatonin, the Authors start the presentation of melatonin by itself.

The whole paragraph is biased, as it is granted in the Authors mind – see table 1 – that not only melatonin is would plenty of things in the body, but it is also, in plants where it can a be – naturally – a source of this natural wonder (we’ll come back to this later on). Let me comment on Table 1:

  1. The ONLY function of melatonin in Mammals (incl. Humans) and other living animals ( all of them?) that is not disputable is the way it directs the circadian rhythm and thus the circannual rhythm
  2. The other functions are not demonstrated, or in very biases way (scientifically). And for reasonable scientists, it is probably at least completely Interestingly, a series of controversies were published, and should be mentioned at one point.
  3. In the plants, the presence – and the roles – of melatonin is another source of .! Indeed: 1/ the way it was measured is not reassuring – many analytical flaws – and the fact that melatonin can save plants from drought and flood, heavy metals toxicity, cold and heat, etc… is also jaw-dropping and very questionable and 2/ the ingestion of melatonin via edible vegetables is also questionable – see Kennaway works.

A good example of the way literature is cited but not understood is the paper by Tan et al (ref 32 or 27??) This paper is an hypothesis completely un-experimental (as often with those authors) and the reality was then experimentally demonstrated later on in J. Pineal research paper.

Reply: We have focused our manuscript on the effects of melatonin on mammals. The actions of the indoleamine in vegetables and plants fall out of the scope of our work. Additionally, we have reviewed the research we could find using searching engines (Pubmed, Google scholar…). We accept the reviewer´s skepticism and we are aware that not all research carried out points in the same direction. We signal at the end of our manuscript that the reader should not consider taking freely melatonin against disease (lines 911-912). In any case, in all the published material the authors will have disclosed any biases or personal/private interests in the outcomes of the studies carried out. There might be some part of truth in the research carried out on this theme worldwide.

QR2/NQO2 is not using melatonin as a substrate or a co-substrate it is barely a 100µM inhibitor of it….

It says: “Additionally, there is at least 400x more melatonin in the gastrointes-150 tinal tract than in the pineal gland [19].” Ref 19 is a review? I am not even sure of the reference number ????.

Please, read it and find the original reports on those organism melatonin concentration. I believed there are none (references). We discussed this major bias elsewhere.

Reply: We have found a study by Sourav Mukherjee, who states that Generally, melatonin concentrations in gut tissues surpass the levels of melatonin in circulation by 10–100 times and gives data about concentrations of the indoleamine in the gastrointestinal parts of different species. This manuscript has been cited (Sourav Mukherjee and Saumen Kumar Maitra. Gut melatonin in vertebrates: chronobiology and physiology. Front. Endocrinol., 22 July 2015. Volume 6 – 2015. https://doi.org/10.3389/fendo.2015.00112). Lines 128-130, reference 24.

3/ Melatonin as an antioxidant

The first sentence of this paragraph states that melatonin is a free-radical scavenger. Presumably, because that can be the case in vivo – or in living systems. Well, HJ Forman works clearly show that scavenger in vivo DOES NOT EXIST. Please, look at those papers and include them in this paragraph. Of course, indoles, like tryptophan, auxin, serotonin, and so on, are scavenger in tubo (in a tube, or in an a-cellular system). Many experiments do show that, but the way they work and its extrapolation into living system is impossible, and not demonstrated.

Then, the confusion between scavenger and inducer of enzymatic defenses is coming in the picture. Yes, melatonin – at very high concentration – induces enzymatic defenses of the cells. But the mechanism behind this induction is unknown – please read the famous and scandalous fraud of the ??? paper. What this fact does say is that induction by melatonin per se is not granted, it might be due to a metabolite, to the indole core or to anything else one does not think about at the moment.

I insist, as I’ll probably repeat later on: concentration is the key, all the magnificent properties of melatonin are reported in models at 100 µM to 10 mM dosage, a laughable dose for anyone concern with drug actions. And certainly not reachable in human (see Harpsoe et al papers, among many others).

The next lines of this paragraph recapitulate the possible targets of melatonin. One should look at the paper of Jockers on the number of targets REPORTED for melatonin (more that 600), and their significance, considering their affinity for the hormone.

Further discussions followed that publication and should be cited in the essay.

Reply: It is expected that what is observed in vitro might also occur in vivo, despite it must be proven. We all should have some optimism in our hands. Bibliography exists that signals that certain substances/compounds/proteins depict in vivo scavenging action against oxidants. Some examples, but not all, support this assumption:

A ROS scavenging protein nanocage for in vitro and in vivo antioxidant treatment. Zhu W, Fang T, Zhang W, Liang A, Zhang H, Zhang ZP, Zhang XE, Li F. Nanoscale. 2021 Mar 4;13(8):4634-4643. doi: 10.1039/d0nr08878a. PMID: 33616146.

Role of vitamin E as a lipid-soluble peroxyl radical scavenger: in vitro and in vivo evidence. Niki E. Free Radic Biol Med. 2014 Jan;66:3-12. doi: 10.1016/j.freeradbiomed.2013.03.022. Epub 2013 Apr 2. PMID: 23557727.

In vitro and in vivo antioxidant activities of polysaccharide purified from aloe vera (Aloe barbadensis) gel. Kang MC, Kim SY, Kim YT, Kim EA, Lee SH, Ko SC, Wijesinghe WA, Samarakoon KW, Kim YS, Cho JH, Jang HS, Jeon YJ. Carbohydr Polym. 2014 Jan;99:365-71. doi: 10.1016/j.carbpol.2013.07.091. Epub 2013 Aug 31. PMID: 24274519.

In vitro and in vivo effects of Laurus nobilis L. leaf extracts. Kaurinovic B, Popovic M, Vlaisavljevic S. Molecules. 2010 May 7;15(5):3378-90. doi: 10.3390/molecules15053378. PMID: 20657487.

Functionality of apigenin as a potent antioxidant with emphasis on bioavailability, metabolism, action mechanism and in vitro and in vivo studies: A review. Kashyap P, Shikha D, Thakur M, Aneja A. J Food Biochem. 2022 Apr;46(4):e13950. doi: 10.1111/jfbc.13950. Epub 2021 Sep 26. PMID: 34569073.

The list of references in this line is huge. If bibliography that has studied the free radical properties of melatonin exists, we humbly consider that the effects and conclusions reported should be accepted. Nevertheless, we agree with the reviewer that there might be certain compounds/substances for which its free radical scavenging properties need to be further proved. We have included in the manuscript reference to the reviewer´s observation and cited studies by HJ Forman. Lines 248-251, “However, controversy exists about the effectiveness in the antioxidant defense of radicals scavenging by nutritional antioxidants. Instead, other mechanisms such as enzymatic removal are the major antioxidant mechanisms [84,85]”.

We have included in the revised version of our manuscript reference to the studies of Nathja Groth Harpsøe and their findings about bioavailability of melatonin, lines 143-146, “However, studies exist which signal that bioavailability of oral melatonin ranged 3-15 % and that melatonin dose should be augmented depending on individual conditions if corresponding melatonin plasma levels are intended [38–40]”.

The next lines of the paragraph refer to the possible targets of melatonin on different organs and tissues from the antioxidant point of view. We have searched the works by Jockers, but we could not find reference in his research to “antioxidant effects” or “antioxidant mechanisms of action” of melatonin.

4/ melatonin and neurological diseases, 5/ melatonin and cancer and 6/ Melatonin and inflammation

As we discussed, not a single clinical study showed melatonin to be active in any of those – numerous – ailments.

Claiming that melatonin is an anticancer agent or adjuvant does not make it real. Claiming that it should be tested in the clinic is a better way to help melatonin case. Citing that it does not work is the way to do it, even if it is “sad”(??).

Reply: As mentioned above, we have given in the text information regarding the fact that melatonin should not be freely taken against disease, despite the scientific evidence that supports encouraging actions of the indoleamine. Moreover, we state that further research is needed to better clarify its putative potential against disease. Conclusion section, lines 922-929: “However, further research is needed to clearly establish the safety of exogenous melatonin for long periods of time. Additionally, at the present, it is difficult to draw any definitive conclusions on the effectiveness of melatonin in reversing aging and restoring, or even prolonging, youth. Despite research seeds some light in this line, scientists still have not found compelling evidence that signals that melatonin supplementation is a valuable tool to prevent disease and/or enlengthen life expectancy. Additional research, involving clinical trials and humans exhibiting long life expectancy, will be need-ed to finally achieve this goal”.

7/ Melatonin and immune diseases.

Same comment than above. ON the top of it, we recently showed that melatonin up to 100µM has no effects on the main immune models.

Reply: Controversy exists, in this and in any other field of research. The information that we have found signals that melatonin may exhibit immunomodulation.

8, 9 / Ageing.

No comments. I found hard to believe that Scientists could be seduced by such millennial ideas (as pointed out in the introduction)

Reply: Thank you very much for your observation.

10/ Conclusions:

At last, the authors express their point of view on the various aspects previously mentioned.

They link the effects of melatonin to antioxidant effects. This is the basis of their reasoning. I am not totally convinced that cancer, antiviral aspects of melatonin actions (while not commented in there), neurological benefits, etc.. can really be linked – at the pathology level – to the prescription of melatonin, but it feels that this is an idea linked to “effective sleep hygiene and implementing habits”.

Line 1413, they wrote “Nevertheless, at the present, it is difficult to draw any definitive conclusions on the effectiveness of melatonin in reversing aging and restoring, or even prolonging, youth.” I guess there is that sentence that summarize our different points of view. The Authors believe. I don’t.

Reply: Thank you very much for your opinion. We have included new text in the conclusion, lines 908-912, “Because the bioavailability of exogenous melatonin as diet supplement is low, it could thus be expected that an adequate rhythm of melatonin secretion in the body appears to be a powerful tool to prevent damage to body tissues and structures. Evidence signals that only supplementation might represent a helping hand when physio-logical release is impaired”. We have removed reference to “effective sleep hygiene and implementing habits” from the text.

Reviewer 3 Report

Comments and Suggestions for Authors

Authors have presented the manuscript entitled Melatonin Interplayinterplay in Physiologyphysiology and Disease. The Fountain of Eternal Youth Revisited. Good topic and excellent literature sources but it needs major revision.

  1. Long introduction with numerous historical facts and songs but nothing about Lerner and his discovery of melatonin.
  2. I recommend to reference melatonin receptors abundance through Pubmed database and some papers, e.g. https://www.sciencedirect.com/science/article/pii/S0753332206000114#aep-bibliography-id26
  3. Please consider controversal effects of melatonin in various cell line models and animal studies.

Author Response

Reply to Reviewer 3 Biomolecules-3549664

Authors have presented the manuscript entitled Melatonin Interplayinterplay in Physiologyphysiology and Disease. The Fountain of Eternal Youth Revisited. Good topic and excellent literature sources but it needs major revision.

Thank you very much for reviewing our manuscript, for the time given to our work and for your suggestions, which we think have helped us improve the quality of our manuscript. Here we send our replies to your comments and suggestions.

  1. Long introduction with numerous historical facts and songs but nothing about Lerner and his discovery of melatonin.

Reply: We have removed the list of historical and artistic quotations from the introduction. Both the abstract and the introduction have been rewritten. Now the introduction is shorter. Following the reviewer’s suggestion, we have mentioned in this section that melatonin was first isolated by Aaron B. Lerner in 1960 from bovine pineal glands. Reference [18] has been given. Lines 109-110.

  1. I recommend to reference melatonin receptors abundance through Pubmed database and some papers, e.g. https://www.sciencedirect.com/science/article/pii/S0753332206000114#aep-bibliography-id26

Reply: Reference to melatonin receptors abundance and distribution was done in the submitted version of the manuscript. We have included in the revised version of our manuscript mentioning the work suggested. Lines 161-162, reference number 48.

  1. Please consider controversal effects of melatonin in various cell line models and animal studies.

Reply: Thank you very much for this comment. We have mentioned in the revised version of our manuscript that the effects of melatonin appear to be cell-type and context-dependent. Whereas it protects healthy cells against noxious agents, cell death is evoked in malignant or transformed cells. Bibliography is cited. Lines 235-240: “However, the effects of melatonin appear to be cell-type and context-dependent. Moreover, because of the ubiquitous nature of melatonin receptors, it functions as a pleiotropic molecule. Furthermore, its multiplicity of action goes beyond the established antioxidant activities. Interestingly, it protects healthy cells against noxious agents, whereas cell death is evoked in malignant or transformed cells [77,78]”.

Reviewer 4 Report

Comments and Suggestions for Authors

Comments and Suggestions for Authors are included in the attached file. 

Author Response

Reply to Reviewer 4 Biomolecules-3549664

The authors provide a comprehensive overview of melatonin's physiological functions and its potential as a protective agent against various diseases. The historical context and cultural references add an engaging element to the introduction. However, the manuscript would benefit from a more focused approach and clearer organization of ideas. Some sections could be condensed, while others may require more in-depth analysis of recent scientific literature.

Thank you very much for reviewing our manuscript, for the time given to our work and for your suggestions, which we think have helped us improve the quality of our manuscript. Here we send our replies to your comments and suggestions.

Questions for the Introduction:

  • Comment: Can the authors provide more recent references to support the claim that melatonin levels are highest in young age and lowest in elderly?
  • Reply: Thank you very much for this comment. New bibliography has been cited and included in the references list. The new references are numbered as 10 to 13.

  • How do the authors justify the inclusion of extensive cultural and historical references in a scientific review? Could this section be condensed to maintain focus on the scientific aspects?

  • Reply: We have removed cultural and historical references from the manuscript.

Questions for the Methods:

  • Did the authors consider any systematic review or meta-analysis methodologies in their approach to synthesizing the literature on melatonin?
  • Reply: Thank you very much for this comment. We will consider preparing a meta-analysis about studies on melatonin actions in a future manuscript.

Questions for the Results:

  • Can the authors provide more quantitative data on melatonin concentrations in various tissues and organs?
  • Reply: Thank you very much for this comment. The bibliography that we have found signals that, in general, it is hardly difficult (or impossible in cases like retina, for example) to measure directly the local secretion of melatonin. We have mentioned this fact in the revised version of our manuscript, and we referenced the data we could find. We have included the following text: “However, it is difficult to determine the exact concentration of melatonin that is locally released at the periphery organs. In general, it is accepted that the concentration of the indoleamine in the periphery is mostly higher than that found in blood”. Lines 122-125.

We have additionally included some new data about melatonin levels in certain locations. Lines 130-137: “In testes, the study by KozioÅ‚ et al. showed that melatonin concentration exhibited varia-tions that depended on the stage of the year. For example, the levels were higher in May (522.50 ± 54.20 pg/mL) compared to July/August (258.50 ± 36.82 pg/mL). During Septem-ber, the melatonin concentration was higher (393.50 ± 36.77 pg/mL) than in July/August but lower than in May [33]. Levels of 20 pg/mL melatonin have been reported in the fluid of small follicles and 10 pg/mL in that of large ones [34]. In bone marrow, concentrations around 413 ± 81 pg/mg have been detected [35]”.

  • How do the effects of endogenous melatonin compare to those of exogenous melatonin supplementation?

  • Reply: Thank you very much for this observation. We have included information regarding this comment in the revised version of our manuscript, lines 220-229. Bibliography has been cited. The text is as follows: “Therefore, it could be conceivable that melatonin supplementation might exert beneficial effects. It is accepted that supplementation with the indoleamine exhibits higher efficacy when endogenous melatonin levels are low. Additionally, most studies suggest that mel-atonin supplementation does not suppress endogenous production even with long-term use [73]. The study by Zhdanova et al. reported increases in serum melatonin levels fol-lowing a low oral dose of the hormone in elderly adults. Nevertheless, the peak reached exhibited variations among people over 48 years old [74]. Moreover, low to moderate dos-ages of melatonin (approximately 5–6 mg daily or less) appears safe thus suggesting that long-term usage might be beneficial to certain patient populations [75]”.

  • What is the current evidence for melatonin's role in specific age-related diseases?
  • Reply: Thank you very much for this comment. Evidence exists in the bibliography, which contains studies referring to impairment of melatonin secretion and age-related diseases. It is well known that melatonin synchronizes central but also peripheral oscillators (fetal adrenal gland, pancreas, liver, kidney, heart, lung, fat, gut, etc.), allowing temporal organization of biological functions through circadian rhythms (24-hour cycles). This is mentioned in the manuscript, lines 197-199.

We have also mentioned in the text the following: “Interestingly, strong reductions of circulating melatonin have been observed in disorders and/or diseases such as Alzheimer’s disease and other neurological and stressful conditions, pain, cardiovascular diseases, cancer, and/or endocrine and metabolic disorders, in particular diabetes type 2 [72]”, lines 217-220.

Thus, it could be expected that the indoleamine exhibits kind of role in disease, i.e., low levels might render the body more prone to appearance of disorders. We have mentioned this in lines 215-216, and bibliography has been included ([71]).

  • Have any clinical trials demonstrated the long-term effects of melatonin supplementation on aging or disease prevention?
  • Reply: The European Union Clinical Trials Register allows us to search for protocol and results information on melatonin use in clinical trials, yielding positive outcomes. Moreover, there is information available which refers to tolerance of exogenous melatonin administration. We have included in the introduction section new information regarding this comment. We state in the text: “Most of the studies have been carried out in vitro. However, in vivo studies also exist, including studies in humans. The European Union Clinical Trials Register allows us to search for protocol and results information on melatonin use in clinical trials, which have yielded positive outcomes. Although most studies are focused on sleep disorders, melatonin has aroused as putative treatment, or at least tool to improve symptoms, for cardiometabolic risk, sepsis, surgery, Covid-19, chronic pain or damage by sun exposition. Role as neuroprotectant in infants and as anxiolytic, or against visual anomalies also have been reported [16]”. Lines 87-95.

Finally, in the Conclusion, we state “Despite research seeds some light in this line, scientists still have not found compelling evidence that signals that melatonin supplementation is a valuable tool to prevent disease and/or enlengthen life expectancy. Additional research, involving clinical trials and humans exhibiting long life expectancy, will be needed to finally achieve this goal”. Lines 925-929.

  • How does melatonin interact with other physiological systems, such as the immune or endocrine systems?
  • Reply: Studies exist that report favorable responses to vaccination in the presence of melatonin. In this line, melatonin may help strengthen the immune system. Information regarding this question is given in the manuscript, lines 694 and following, in the section dedicated to inflammation and immunity.

Additionally, melatonin might act on endocrine organs, like for example the adrenal glands, where it might act as an endogenous pacemaker. This might be related to the oscillation of glucocorticoids levels between night and day. We have focused our manuscript onto the effects of melatonin against diseases related to aging. We think that interactions of melatonin with the endocrine system fall out of the topic reviewed.

Questions for the Discussion:

  • What are the potential limitations or risks associated with long-term melatonin use?

Reply: The information available in the bibliography reports that melatonin is safe to use long-term. Melatonin at low to moderate dosages (approximately 5–6 mg daily or less) appears safe. Long-term usage appears to benefit certain patient populations. However, the long-term effects of taking exogenous melatonin have been insufficiently studied and require additional investigation. Reference for this cooment has been included in the text, lines 227-229 (with bibliography, nº 75) and in the conclusion section, lines 921-923, respectively.

  • How do the authors envision translating the findings on melatonin's protective effects into practical clinical applications?
  • Reply: The employment of melatonin in the treatment of certain disorders, especially those related to sleep, has been carried out for a long time, yielding positive results. Moreover, as mentioned above, clinical trials have been developed, not only for the treatment of sleep disorders but also for the treatment of neurological and cardiac diseases, to cite some examples. Therefore, the results of in vitro studies have already encouraged clinicians to test the effects of melatonin on human beings to treat disease.

  • What future research directions do the authors suggest to further elucidate melatonin's role as a potential "fountain of youth"?

Reply: In our humble opinion, the research should be directed to deeply analyze the effects of melatonin on DNA repair systems, and on development of malignancy and senescence. Additionally, the conditions that yield higher plasma levels of the indoleamine in a certain population of elderly people, that leads them live longer, in comparison to others need to be investigated.

Round 2

Reviewer 1 Report

Comments and Suggestions for Authors

After the revision of the article, there has been great progress and improvement, but before accepting it, the full text needs to be reviewed and revised to modify the language and wording, and some English grammar and wording should be done.

Author Response

Reply to Reviewer 1 Biomolecules-3549664

Comments and Suggestions for Authors

After the revision of the article, there has been great progress and improvement, but before accepting it, the full text needs to be reviewed and revised to modify the language and wording, and some English grammar and wording should be done.

Thank you very much for reviewing our manuscript, for the time given to our work and for your suggestions, which we think have helped us improve the quality of our manuscript. Here we send our replies to your comments and suggestions.

Comment: The full text needs to be reviewed and revised to modify the language and wording, and some English grammar and wording should be done.

Reply: Following the reviewer’s comment we have checked the text for grammar and wording corrections.

Reviewer 2 Report

Comments and Suggestions for Authors

No further comments.

We "agree" to disagree.

Author Response

Reply to Reviewer 2 Biomolecules-3549664

Comments and Suggestions for Authors

No further comments.

We "agree" to disagree.

Thank you very much for reviewing our manuscript, and for the time and effort given to our work.

Reviewer 4 Report

Comments and Suggestions for Authors

 The authors have adequately addressed all the concerns and suggestions raised in the previous review. The methodology is now clearly explained, the results are presented in a more coherent manner, and the discussion provides appropriate context for the findings. The revised manuscript demonstrates significant improvement in both content and clarity, making it suitable for publication in this journal. I recommend this manuscript for publication in its current form.

Author Response

Reply to Reviewer 4 Biomolecules-3549664

Comments and Suggestions for Authors

The authors have adequately addressed all the concerns and suggestions raised in the previous review. The methodology is now clearly explained, the results are presented in a more coherent manner, and the discussion provides appropriate context for the findings. The revised manuscript demonstrates significant improvement in both content and clarity, making it suitable for publication in this journal. I recommend this manuscript for publication in its current form.

Thank you very much for reviewing our manuscript, and for the time and effort given to our work.
